# A unified neural account of contextual and individual differences in altruism

Jie Hu[1]*, Arkady Konovalov[1,2], Christian C Ruff[1,3]*

[1]Zurich Center for Neuroeconomics, Department of Economics, University of Zurich, Zurich, Switzerland; [2]Centre for Human Brain Health, School of Psychology, University of Birmingham, Birmingham, United Kingdom; [3]University Research Priority Program 'Adaptive Brain Circuits in Development and Learning' (URPP AdaBD), University of Zurich, Zurich, Switzerland

**Abstract** Altruism is critical for cooperation and productivity in human societies but is known to vary strongly across contexts and individuals. The origin of these differences is largely unknown, but may in principle reflect variations in different neurocognitive processes that temporally unfold during altruistic decision making (ranging from initial perceptual processing via value computations to final integrative choice mechanisms). Here, we elucidate the neural origins of individual and contextual differences in altruism by examining altruistic choices in different inequality contexts with computational modeling and electroencephalography (EEG). Our results show that across all contexts and individuals, wealth distribution choices recruit a similar late decision process evident in model-predicted evidence accumulation signals over parietal regions. Contextual and individual differences in behavior related instead to initial processing of stimulus-locked inequality-related value information in centroparietal and centrofrontal sensors, as well as to gamma-band synchronization of these value-related signals with parietal response-locked evidence-accumulation signals. Our findings suggest separable biological bases for individual and contextual differences in altruism that relate to differences in the initial processing of choice-relevant information.

**\*For correspondence:**
hujie0223@gmail.com (JH);
christian.ruff@econ.uzh.ch (CCR)

**Competing interest:** The authors declare that no competing interests exist.

## Editor's evaluation

In this important paper, the authors use a sophisticated combination of computational modeling and EEG to show that variation in generosity produced by changes in context (i.e., disadvantageous vs. advantageous inequality) and variation due to individual differences in concern for others both seem to occur early, during the perceptual or valuation stage of a choice, rather than later on during choice comparison. However, these two sources of variation also appear to operate through distinct mechanisms during this stage of processing, which spurs further questions about the drivers of human prosocial behavior. This paper will be of considerable interest to researchers studying the psychological and neural basis of variation in prosocial behavior.

## Introduction

Altruism – incurring own costs to benefit others – is fundamental for cooperation and productivity in human societies (*de Waal, 2008*; *Piliavin and Charng, 1990*). It not only plays crucial roles in shaping socio-political ideology and welfare (e.g. via tax policies and charity; *Bechtel et al., 2018*; *Offer and Pinker, 2017*) but is also essential for collective management of challenging situations, such as political, financial, and public health crises. While altruism is thought to be a stable behavioral tendency shaped by the evolutionary advantages of the ability to cooperate, it is unclear why this tendency varies so strongly across individuals, contexts, and cultures (*Bester and Güth, 1998*; *Hamilton, 1964a*;

*Hamilton, 1964b*; *Lebow, 2018*; *Piliavin and Charng, 1990*). Is altruism governed by a set of unitary neuro-cognitive mechanisms that are engaged to varying degrees in different situations or different people (*Tricomi et al., 2010*)? Or are there fundamentally different types of altruistic actions that are guided by different neuro-cognitive processes triggered by different contexts (*Hein et al., 2016*)?

From a neurobiological perspective, both these possibilities appear plausible. On the one hand, all altruistic actions necessitate the ability to override self-interest, a parsimonious brain mechanism (*Bester and Güth, 1998*) that is thought to be facilitated more or less by different contexts and that could be expressed to different degrees in different people (*Morishima et al., 2012*; *Trivers, 1971*). On the other hand, empirical observations suggest that altruism varies with a range of factors such as others' previous actions (e.g. empathy-based vs. reciprocity-based altruism) or their perceived similarity (e.g. social distance; *Hein et al., 2016*; *Vekaria et al., 2017*). It is thus often argued that in different contexts or different individuals, superficially similar altruistic actions can be guided by distinct motives (such as personal moral norms, responsibility, or empathy), which may be controlled by fundamentally different types of neurocognitive mechanisms (*Hein et al., 2016*; *Piliavin and Charng, 1990*; *Zaki and Mitchell, 2011*).

One specific context factor that is often discussed in this context is the inequality in resources held by the actor and the recipient of a possible distribution: People are more willing to share if they possess more than the recipient (advantageous inequality, ADV) than if they possess less (disadvantageous inequality; DIS) (*Charness and Rabin, 2002*; *Fehr and Schmidt, 1999*; *Gao et al., 2018*; *Güroğlu et al., 2014*; *Morishima et al., 2012*; *Tricomi et al., 2010*). Although this consistent effect has been formalized with the same utility model across contexts, this model needs to comprise two distinct latent parameters quantifying altruism in the two contexts (i.e. decision weights on others' payoffs that are specific for ADV and DIS), and these are often uncorrelated and differ strongly from each other (*Gao et al., 2018*; *Morishima et al., 2012*). These observations, together with distinct psychological accounts for the distribution behaviors in different contexts (i.e. 'guilt' in the advantageous and 'envy' in the disadvantageous inequality context), imply that altruistic choices in the two contexts may be driven by fundamentally different psychological processes (*Fehr and Schmidt, 1999*; *Gao et al., 2018*). Moreover, modeling studies often reveal that these altruism parameters vary strongly between different people for the same choice set (*Fehr and Schmidt, 1999*), and neuroimaging studies have shown that while distributional behavior in both contexts correlates with activity in brain regions commonly associated with motivation (e.g. the putamen and orbitofrontal cortex), either context also leads to activity in a set of distinct areas (the dorsolateral and dorsomedial prefrontal cortex in advantageous and the amygdala and anterior cingulate cortex in disadvantageous inequality; *Gao et al., 2018*; *Yu et al., 2014*). Finally, neuroanatomical research shows that only for advantageous inequality, individual variations in altruistic preferences relate to gray matter volume in the temporoparietal junction (TPJ; *Morishima et al., 2012*).

While these behavioral modeling and neural findings suggest clear contextual and individual differences in altruism, it is still unclear what specific neurocognitive mechanisms these differences could arise from. Previous research on individual and contextual differences in altruism has largely used unitary computational models focusing exclusively on valuation (rather than attempting to separate distinct aspects of the choice process), and has used functional magnetic resonance imaging (fMRI) to identify spatial patterns of neural activity that correlate with valuation processes during wealth distribution behaviors in different contexts (*Charness and Rabin, 2002*; *Fehr and Schmidt, 1999*; *Gao et al., 2018*; *Güroğlu et al., 2014*; *Morishima et al., 2012*; *Tricomi et al., 2010*). For example, recent studies combined computational modeling with fMRI techniques to show that the value of altruistic choice can be modeled as the weighted sum of self- and other-interest, and that different attributes are integrated into an overall value signal correlating with BOLD activity in the ventromedial prefrontal cortex (vmPFC) (*Crockett et al., 2017*; *Crockett et al., 2013*; *Hutcherson et al., 2015*; *Hare et al., 2010*). However, since these studies neither formally examined the difference in altruistic choices between advantageous and disadvantageous inequality contexts, nor focused on separating different aspects of the decision mechanisms of altruistic choice, they can hardly address the question of whether and how different mechanisms are involved in different types of altruistic actions in different contexts (*Crockett et al., 2013*; *Crockett et al., 2008*; *Gao et al., 2018*).

To systematically investigate this issue, it would be beneficial to harness the fact that altruistic decisions – like all choices – are guided by processes unfolding at different temporal stages (*Seo and*

*Lee, 2012*; *Shin et al., 2021*; *Tump et al., 2020*). These processes include (1) initial perception of the objective information related to wealth distribution (e.g. payoff numbers) (*Nieder, 2016*; *Pinel et al., 2004*), (2) biased representations of the subjectively decision-relevant information attributes, such as attention-guided weighing of self- vs other-payoffs (*Chen and Krajbich, 2018*; *Teoh et al., 2020*), (3) integration of all these attributes and subjective preferences into decision values (*Collins and Frank, 2018*; *Harris et al., 2018*; *Hutcherson et al., 2015*), and (4) final decision processes that transform the decision values into motor responses (*O'Connell et al., 2012*; *Polanía et al., 2014*). Taking into account this temporal unfolding of the neurocognitive processes further refines the questions about the origins of differences in altruistic behavior: Do altruistic choices involve different sets of computations throughout all the temporally different processing stages (i.e., initial perceptual processing, valuation, final integrative choice mechanisms) in these different contexts and by different individuals (as suggested by *Gao et al., 2018*; *Tricomi et al., 2010*)? Or do individuals mainly perceive and attend to the choice-relevant information differently, before passing on this information to valuation and integrative decision mechanisms devoted to all types of altruistic choices (as suggested by *Yu et al., 2014*)? Answering these questions by means of modelling and neural recording techniques that allow a detailed focus on different temporal stages of altruistic choice processes could help us understand the biological origins of altruism, reveal why people differ strongly in altruistic behavior, and develop more efficient strategies to facilitate altruism.

In the current study, we take such an approach. We combined a modified dictator game that independently varies payoffs to a player versus another person, and thereby also the inequality between both players, with electroencephalography (EEG) and sequential sampling modeling (SSM). This allowed us to identify electrophysiological markers of the initial perceptual processing and biased representation of the decision-relevant information (i.e. stimulus-locked event-related potentials [ERPs] related to the payoffs and the inequality context) as well as of the processes integrating this information into a decision variable used to guide choice (i.e. response-locked evidence accumulation [EA] signals; *Balsdon et al., 2021*; *Hutcherson et al., 2015*; *Krajbich et al., 2015*; *Nassar et al., 2019*). Thus, our approach differs from that of fMRI studies identifying brain areas involved in the valuation of own and others' payoffs (*Fehr and Schmidt, 1999*; *Morishima et al., 2012*; *Sáez et al., 2015*), since the temporal resolution of fMRI measures makes it difficult to separate response-locked decision-making processes from stimulus-locked perceptual processes and to examine the independent dynamics of these processes during distribution decisions.

Our approach is also motivated by studies of nonsocial decisions showing that SSMs may provide a useful framework for investigating the temporal dynamics of the processes that integrate different choice attributes into the decision outcome (*Harris et al., 2018*; *Maier et al., 2020*). Many studies have shown that SSMs can identify these processes not just computationally, but also at the neural level, for both the perceptual (*Brunton et al., 2013*; *Kelly and O'Connell, 2013*; *Ossmy et al., 2013*) and value-based decision making (*Glaze et al., 2015*; *Hutcherson et al., 2015*; *Pisauro et al., 2017*; *Polanía et al., 2014*). The SSM framework provides a formal way to predict the temporal dynamics of processes that integrate evidence for one choice option over another for the temporal period leading up until choice, and to separate these from initial perceptual processes time-locked to stimulus presentation. Neural signals corresponding to these predicted evidence-accumulation signals have been identified with EEG for perceptual decision making across different sensory modalities or stimulus features (*Kelly and O'Connell, 2013*; *O'Connell et al., 2012*; *Wyart et al., 2012*) as well as for value-based decision making (*Pisauro et al., 2017*; *Polanía et al., 2014*). These studies have identified evidence accumulation processes either as the model-free build-up rate of the centroparietal positivity (CPP) (*Kelly and O'Connell, 2013*; *Loughnane et al., 2018*; *Loughnane et al., 2016*; *O'Connell et al., 2012*) or in SSM-prediction-based neural signals measured over parietal and/or frontal regions (*Pisauro et al., 2017*; *Polanía et al., 2014*). Both types of neural signals are commonly interpreted as reflecting integration of the choice-relevant evidence to reach a decision, rather than basic motor planning which is usually identified by a fundamentally different neural signal, the contralateral action readiness potential (*Kornhuber and Deecke, 2016*; *Schurger et al., 2021*). The cortical origins of these signals may in principle correspond to locations identified by fMRI studies of corresponding SSM-predicted evidence accumulation traces, but note that these studies were not able to study the temporal dynamics of such signals and to unambiguously separate them into stimulus-locked

perceptual versus response-locked decision processes (*Gluth et al., 2012*; *Hare et al., 2011*; *Hutcherson et al., 2015*; *Rodriguez et al., 2015*).

Studies using this approach to investigate different types of decisions have identified different cortical areas that implement evidence-accumulation signals in different choice contexts (e.g. parietal regions specifically for perceptual decision making vs. both frontal and parietal regions for value-based decision making *Polanía et al., 2014*). This shows that different types of decisions may, even if they are reported via the same manual actions, draw on evidence accumulation computations that are instatiated in distinct brain regions. Moreover, altruistic decisions driven by different motives, or made by individuals with different social preferences, have also been found to involve activity in different neural networks (*Hein et al., 2016*). Therefore, it is necessary to differentiate whether the contextual and individual differences in altruistic decisions reflect recruitment of different brain areas/ signals and/or of different computations that are performed within these brain areas. If different final decision mechanisms (i.e. computational and/or neural mechanisms) were to be involved in the two types of altruistic choices, or in different individuals, we should observe response-locked evidence-accumulation signals in different brain areas (e.g. frontal vs. parietal regions), or even different types of computations, in the two types of inequality contexts and/or different individuals. Conversely, if the same final decision mechanism is employed for both types of choice contexts, we should observe similar evidence-accumulation neural signals in similar brain areas, but systematic variations across contexts and/or individuals in those signals (e.g. responses in different brain areas and/or with different temporal characteristics) related to early perceptual/attentional processing of choice-relevant information, such as the available payoff magnitudes (*Harris et al., 2018*).

Here, we apply this approach and use SSMs fitted to individuals' wealth distribution behaviors to predict the underlying neural evidence accumulation dynamics. We then employ these predicted EA signals in our EEG analyses to examine whether a similar neural choice system accumulates the choice-relevant evidence in both inequality contexts, or whether distinct neural systems implement this decision process for the different contexts. Then, we examine whether the different features of each choice problem that ultimately need to be integrated into the choice-relevant evidence – that is, the specific payoffs available to oneself and the other person – are initially processed in a different manner for different contexts and in different individuals. This allows us to directly approach the question of whether contextual and individual differences in altruism arise from differences in the decision mechanisms that integrate and compare choice-relevant information at the final stage of the choice process, or rather from differences in the initial processing and biased representation of the choice-relevant information that is ultimately integrated into the final decision mechanism.

## Results

We recorded 128-channel EEG data from healthy participants playing a modified Dictator Game (DG). On each trial of this task, participants played as proposers and chose between two possible allocations of monetary tokens between themselves and an unknown partner. We systematically varied the allocation options from trial to trial so that in half of the trials, participants received less than their partners for both choice options (disadvantageous context [DIS]) and in the other half they got more than their partners for both options (advantageous context [ADV]). These two types of trials were randomly intermixed and were only defined by the size of the payoffs presented on the screen.

On each trial, we presented the two options sequentially, to allow clear identification of time points at which the information associated with each option was processed (*Figure 1A*, see Materials and methods for details). This sequential presentation allowed us to establish the inequality context with the presentation of the first option, without having to explicitly instruct participants about the two contexts. We then studied individuals' sensitivity to self-payoff and other-payoff by focusing on how the choice of the second option depended on the change in these variables from the first to the second option. Importantly, as shown in the payoff schedule of all trials (*Figure 1—figure supplement 1*), we matched self-/other-payoff differences and the resulting absolute levels of inequality across both contexts and also across the second and the first options (*Figure 1—figure supplement 1* middle and right panels). This allowed us to compare choices and response times, model-defined neural choice processes time-locked to the response, and neural processing of different stimulus information (self- and other-payoff) between the two contexts.

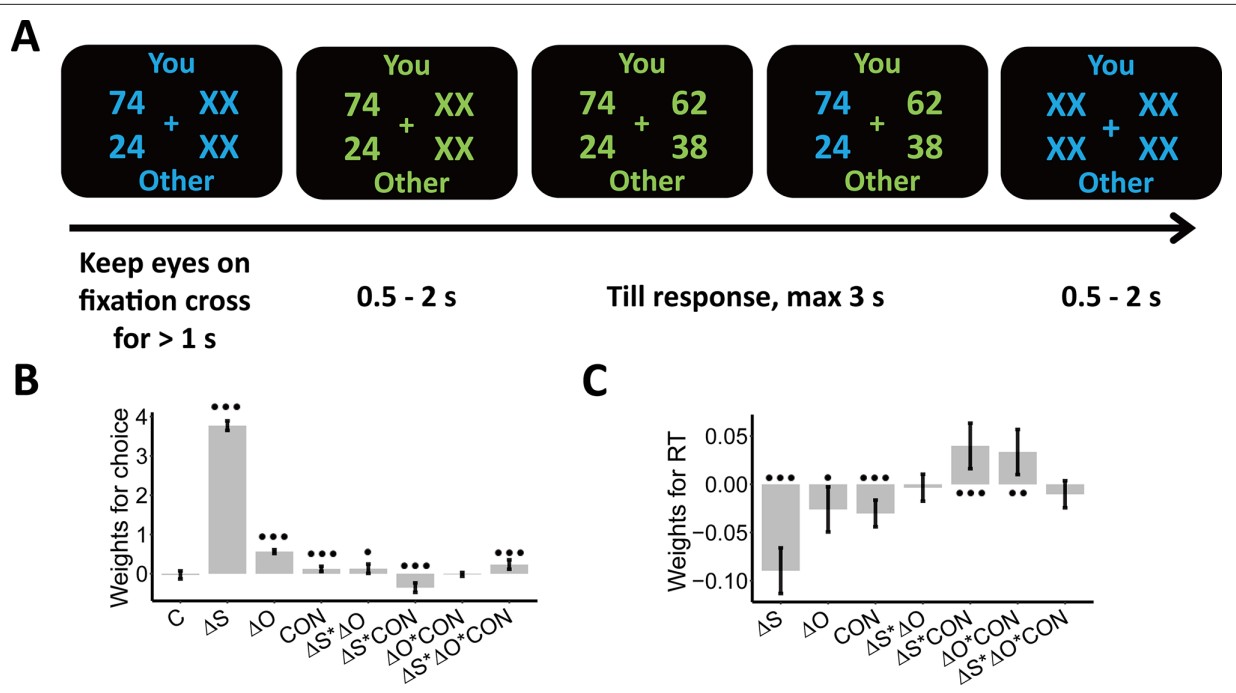

**Figure 1.** Experimental design and behavioral results. We employed a modified dictator game to measure individuals' wealth distribution behaviors. (**A**) Example of display in a single trial. In the task, participants played as proposers to allocate a certain amount of monetary tokens between themselves and anonymous partners. At the beginning of each trial, participants were presented with one reference option in blue and were asked to keep their eyes on the central cross for at least 1 s to start the trial, as indicated by the change in font color from blue to green. When the second option was presented, participants had to choose between the two options within 3 s. The selected option was highlighted in blue before the inter-trial interval. Font color assignment to phases (i.e. blue and green to response) was counterbalanced across participants. (**B**) Payoff information and context affect choice systematically. The generalized linear mixed-effects model shows the effects of multiple predictors on the probability to choose the second option; (**C**) Payoff information and context affect response times systematically. The linear mixed-effects model shows the effects of multiple predictors on response times (RTs). $\Delta S$, Self-payoff Change; $\Delta O$, Other-payoff Change; CON, Context; C, Constant; •••, $p < 0.001$; ••, $p < 0.01$; •, $p < 0.05$. Error bars indicate 95% confidence interval (CI) of the estimates, N=38.

The online version of this article includes the following figure supplement(s) for figure 1:

**Figure supplement 1.** Payoff schedule.

**Figure supplement 2.** Model-free behavioral results.

Based on the model fits and their predicted response-locked evidence accumulation EEG traces, we first tested whether similar or different neural processes (i.e. brain regions or physiological markers) underlie the ultimate choice process in the two inequality contexts, in similarity to how this has been studied for other types of decisions (*Polanía et al., 2014*). Then, we clarified whether neural processing of the stimulus information – which subsequently feeds into the decision processes – differs across contexts and individuals. For this analysis, we examined stimulus-locked event-related potentials (ERPs), in a way that has also been used to differentiate neural processing of decision-relevant features in non-social value-based decision making (e.g. perceptions of health and taste of food items) (*Harris et al., 2018*). Finally, we explored how individual differences in altruism are related to large-scale information communications between regions associated with these two sets of processes (i.e. response-locked decision processes and stimulus-locked perceptual processes), by examining inter-regional synchronization in the gamma-band frequency (30–90 Hz). This last analysis was motivated by the consideration that evidence accumulation processes need to integrate evidence input from different neural sources (e.g. perceptual processes) (*Polanía et al., 2014*), and by the proposal that coherent phase-coupling in the gamma band between different groups of neurons may serve as a fundamental process of neural communication for information transmission (*Bosman et al., 2014*; *Fries, 2009*; *Fries, 2005*; *Vinck et al., 2013*), as already shown for non-social value-based decisions (*Polanía et al., 2014*; *Siegel et al., 2008*).

## Behavior: Altruism depends differentially on self- and other-payoffs across contexts

Before performing model-based analyses, we ran model-free linear mixed-effects regressions to establish that the choice-relevant information (i.e. self-payoff, other-payoff, and inequality context [ADV and DIS]) indeed systematically affects individual wealth distribution choices. These analyses confirmed that both self-payoff and other-payoff were important factors underlying individuals' choices.

Specifically, participants chose the second option more often when either they or the receiver profited more from this choice (main effect Self-payoff Change ($\Delta$S): beta = 3.77, 95% CI [3.65–3.89], p < 0.001; main effect Other-payoff Change ($\Delta$O): beta = 0.56, 95% CI [0.51–0.61], p < 0.001, $\Delta$S($\Delta$O): participants' own (partners') payoff change between the second and the first option) (*Supplementary file 1*, *Figure 1B*). However, participants were less influenced by changes in their own payoff when they had more money than the other (ADV, interaction Self-payoff Change ($\Delta$S) and Context (CON): beta = –0.35, 95% CI [-0.47 to –0.23], p < 0.001) or when the receiver got lower payoffs from this choice (decreasing other-payoff, interaction Self-payoff Change ($\Delta$S) and Other-payoff Change ($\Delta$O): beta = 0.13, 95% CI [0.01–0.24], p = 0.03). This latter effect was particularly marked when the participants had more money than the receiver (ADV; three-way interaction Self-payoff Change ($\Delta$S), Other-payoff Change ($\Delta$O), and Context (CON): beta = 0.23, 95% CI [0.11–0.35], p < 0.001; *Supplementary file 1*, *Figure 1B*). For visualizations of these effects, see Appendix 1 and *Figure 1—figure supplement 2A*, for confirmation by model-based analyses reported, see Appendix 1. Note that we also constructed simpler models without interaction effects and/or main effects, but model comparison analyses favored the full model (*Supplementary file 1*). An additional linear mixed-effects regression model suggested that the presentation order (i.e. first or second) of options would not affect individuals' equal/unequal choices (see Appendix 1 and *Supplementary file 2*).

Self-payoff, other-payoff, and context also jointly affected how quickly participants took their decisions. Choices were faster for larger absolute values of self-payoff change (main effect Self-payoff Change (l$\Delta$Sl): beta = –0.09, 95% CI [-0.11 to –0.07], p < 0.001) and other-payoff change (main effect of Other-payoff Change (l$\Delta$Ol): beta = –0.03, 95% CI [-0.05 to –0.003], p = 0.03) (*Figure 1C*). Again, both these effects were different for the two inequality contexts, with response times more strongly affected in the disadvantageous inequality context (interaction between Self-payoff Change (l$\Delta$Sl) and Context (CON): beta = 0.04, 95% CI [0.02–0.06], p < 0.001; interaction between Other-payoff Change (l$\Delta$Ol) and Context (CON): beta = 0.03, 95% CI [0.01–0.06], p = 0.005; *Supplementary file 3*, *Figure 1C*). These effects are consistent with the central assumption of the SSM framework that stronger (weaker) evidence will speed up (slow down) evidence accumulation and resulting choice, thereby already suggesting that an SSM-based decision process may integrate self- and other-payoff to guide individual decisions (For visualizations of these effects, see *Figure 1—figure supplement 2B*).

## Model-based EEG reveals similar parietal evidence accumulation across contexts

To address the question of whether distribution choices are supported by similar or different neural decision processes across both inequality contexts, we fitted a dynamical sequential sampling model (SSM) to participants' behavioral data and used it to predict neural evidence accumulation (EA) signals for the two contexts.

Our analyses revealed comparable SSM-based EA signals over similar parietal regions for both contexts and no context-specific EA signals that would indicate the use of fundamentally different final choice mechanisms in the different contexts. Specifically, we first fitted the SSM by categorizing trials as 'equal' or 'unequal' choices, based on whether the participant selected the option with more equal or less equal distribution of monetary tokens between both players. For each trial, the model used the subjective value difference (VD) between the more equal option and the more unequal option (computed using the Charness-Rabin utility model, see Materials and methods) as its input to predict moment-by-moment evidence accumulation signals until the timepoint when the decision was made. For this, we used the Ornstein-Uhlenbeck choice model (OU), which assumes a leaky accumulation-to-bound process (*Bogacz et al., 2006*):

$$EA_{(t+1)} = EA_{(t)} + \left( \lambda_{(c,s)} \times EA_{(t)} + \kappa_{(c,s)} \times VD_{(c,s,i)} \right) dt + N\left(0, \sigma\right) \tag{1}$$

$$VD_{(c,s,i)} = \left( \left(1 - \omega_{(c,s)}\right) \times E^S_{(c,s,i)} + \omega_{(c,s)} \times E^O_{(c,s,i)} \right) - \left( \left(1 - \omega_{(c,s)}\right) \times I^S_{(c,s,i)} + \omega_{(c,s)} \times I^O_{(c,s,i)} \right) \tag{2}$$

with indices c for conditions (c = DIS for disadvantageous inequality context, c = ADV for advantageous inequality context), s for participants (s = 1,..., $N_{participants}$), and i for trials (i = 1,..., $N_{trials}$). $E^S_{(c,s,i)}$ $\left(I^S_{(c,s,i)}\right)$ indicates participants' payoff of the equal (unequal) option in condition c, for participant s and trial i; $E^O_{(c,s,i)}$ $\left(I^O_{(c,s,i)}\right)$ indicates the partners' payoff in the equal (unequal) option in condition c, for participant s and trial i.

This model allowed us to fit several free parameters that correspond to different aspects of preference and the choice process: the relative decision weight on others $\omega$ (altruistic preference), decision threshold $\alpha$ (response caution), starting point $\beta$ (response bias), and drift rate modulator $\kappa$ (sensitivity to equality-related information), as well as parameters which are less plausible to be linked to the cognitive or neural mechanisms underlying valuation or decision processes, including leak strength $\lambda$ and non-decision time $\tau$ (see Materials and methods for a detailed model description). By including these parameters, we could examine the effects of context on both basic altruistic preference (i.e. $\omega$) and the final decision process that integrates the subjective values passed on from perception and valuation processes (i.e. $\alpha$, $\beta$, and $\kappa$). Although the payoff of each option was fixed for each trial, participants still had to accumulate evidence by calculating and comparing the difference in payoffs between options, so the decision time limit (3 s) may have accelerated the evidence accumulation speed when the decision process approached the limit. The leak strength parameter $\lambda$ thus captured how the decision process adaptively controls the acceleration or deceleration of evidence accumulation. Please see Materials and methods for the comparison of alternative models and the details of the best-fitting model we used for our analysis.

To simulate the evidence accumulation process, we averaged 500 EA traces generated by the participant-specific fitted model for the given context and the payoffs on each trial. Model simulations showed that these traces were good approximations of the EA processes underlying choice, since the fitted model could both capture choices and RTs across the two contexts. For both types of choices (equal/unequal) and contexts (ADV and DIS), the sensitivity/specificity of the data simulated by the model was higher than 83% (*Figure 2A* left panel) and the balanced accuracy was higher than 89% (*Figure 2A* right panel, Materials and methods). The model also correctly captured response speed effects, predicting that choices are faster during advantageous inequality overall (RT in ADV: 0.80±0.04 s, DIS: 0.84±0.04 s, ADV vs. DIS: 95% CI [-0.06,–0.02], Cohen's d = –0.83, t(37) = –5.12, p < 0.001), and faster for equal (unequal) choices in DIS (ADV) (DIS: RT equal choice: 0.78±0.03 s, unequal choice: 0.91±0.04 s; equal vs unequal choice: 95% CI [-0.16,–0.10], Cohen's d = –1.51, t(37) = –9.31, p < 0.001; ADV: RT unequal choice: 0.73±0.03 s, equal choice: 0.87±0.04 s; unequal vs equal choice: 95% CI [-0.17,–0.11], Cohen's d = –1.54, t(37) = –9.47, p < 0.001, *Figure 2B*).

To identify the corresponding neural traces, we performed cross-validated regressions of EEG signals on the SSM-based EA predictions. We divided trials in each context into even- and odd-numbered trials and used the former to identify channels in which the ERP magnitude correlated with the model-predicted EA traces. We then used regressions to test the model predictions for the data extracted from these channels for the independent odd-numbered trials (see Materials and methods for details). In both contexts, we observed model-predicted neural EA signals in the same channels located over parietal regions (*Figure 2C and D* leftmost and middle-left panels, $p_{Bonferroni}$ < 0.05), and ERP waveforms extracted from these channels for the independent odd-numbered trials showed a significant correspondence with the model-predicted EA signals (DIS: $R^2$ = 0.82, p < 0.001; ADV: $R^2$ = 0.84, p < 0.001, *Figure 2C and D* middle-right panel). These neural expressions of EA signals did not differ between advantageous and disadvantageous inequality contexts, as ascertained by in-depth comparisons of the observed ERP data (t-tests for each EEG electrode did not reveal any significant differences, see Materials and methods for detailed descriptions of statistics).

We also confirmed that decision signals were comparable across contexts using an alternative model-free analysis approach established in the literature (*Kelly and O'Connell, 2013*). This analysis calculates the build-up rates of the response-locked ERP waveforms in the clusters shown in *Figure 2C–D* and correlates this rate with the drift rate modulator derived from the fitted SSM for

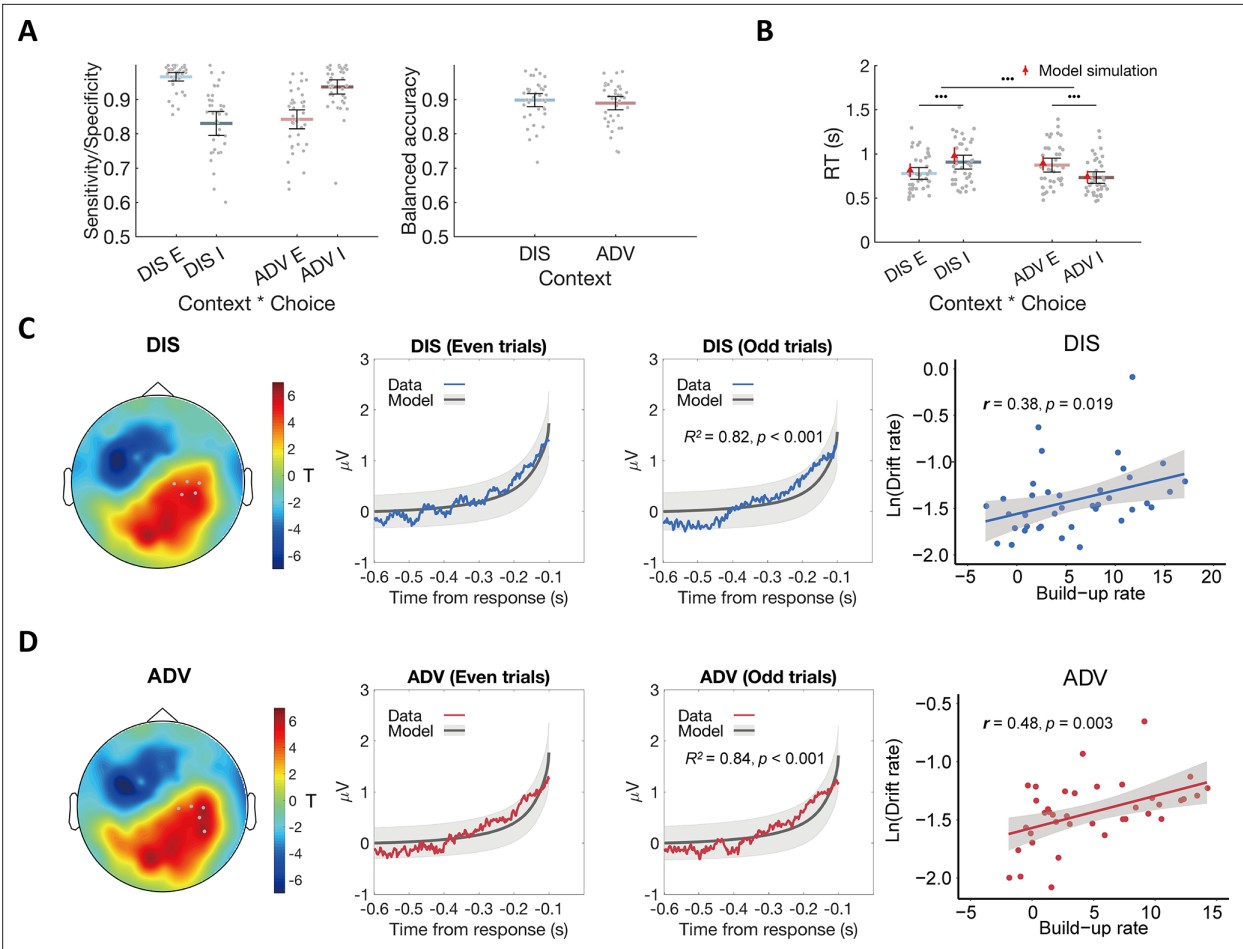

**Figure 2.** Model fits and the relationship between the ERP signal and model predictions. (**A**) The OU model predicts choices across contexts. Left panel: Model performance on sensitivity of equal choice (**E**) and specificity of unequal choice (**I**) in each context. Right panel: Model performance on balanced accuracy in each context. DIS, disadvantageous context; ADV, advantageous context; E, equal choice; I, unequal choice. (**B**) The OU model recovers RT effects over context and choice in participants' behavioral data. Black error bars display means ± 95% confidence intervals (CIs), N = 38. Each grey dot indicates one participant. The red triangle dots and error bars represent the model simulation mean and 95% CIs. •••, p < 0.001. (**C**) & (**D**) Comparable neural evidence accumulation signals in both contexts. For each context (C for DIS, D for ADV), response-locked epochs were divided into even and odd trials to perform cross-validation analyses. Leftmost panels show the topographic scalp distributions of associations between ERP amplitude and OU model predictions in even trials. Grey dots highlight channels that survived the threshold. Middle-left panels show the observed averaged ERP data in the identified clusters shown in leftmost panels (colored lines) and normalized OU model predictions for even-numbered trials. Middle-right panels show the relationships between the model predictions and ERP signals of the independent half of data (odd-numbered trials) in the identified clusters shown in the leftmost panels. Colored lines represent average ERP data extracted from the identified clusters. Grey lines represent the means of model-predicted EA traces, and grey shaded areas represent ± 1 standard error of the mean (SEM) of the model-predicted EA traces, N = 38. Rightmost panels show the correlations between the built-up rates of the identified clusters and the OU model parameter of drift rates across participants.

The online version of this article includes the following figure supplement(s) for figure 2:

**Figure supplement 1.** ERP amplitude at response across inequality contexts.

each context. ERP build-up rates were indeed associated with the drift rates in both inequality contexts (DIS: *r*(37) = 0.38, p = 0.019; ADV: *r*(37) = 0.48, p = 0.003) (*Figure 2C and D* rightmost panels), also supporting the conclusion that similar brain areas implement evidence-accumulation processes for both inequality contexts. Moreover, the ERP amplitude in these sensors in the time window of 100–0ms before the response was greater in the DIS (1.91±0.25) compared to the ADV context (1.70±0.22; DIS vs ADV: 95% CI [0.01, 0.42], Cohen's d = 0.33, *t*(37) = 2.12, p = 0.04). This is entirely consistent with the modelling result that the decision threshold was higher in the DIS compared to the ADV context (*Figure 2—figure supplement 1*). Together, our findings suggest that altruistic decisions in the two contexts recruited the same computations implemented in the same brain areas, and the convergence

of the results from both analysis approaches highlights the similarity of the signals observed in our data to those observed in the same parietal sensors for other types of decision making, such as perceptual and non-social value-based choices (*Pisauro et al., 2017*; *Polanía et al., 2014*).

## Contextual differences in aspects of unified decision mechanism

Given the EEG evidence for a comparable neural decision process across both inequality contexts, we next examined which specific aspects of information processing may vary to produce behavioral differences between the two inequality contexts. We focused on four decision-relevant parameters, including the relative decision weight $\omega$ capturing perception of other-payoff versus self-payoff (i.e. altruistic preferences), the decision threshold $\alpha$ indexing response caution, the starting point $\beta$ quantifying response bias, and the drift rate $\kappa$ measuring the sensitivity of evidence integration (see *Figure 3—figure supplement 1* for the distributions of response urgency with increasing time (leak strength $\lambda$) and speed of pre- and post-decision information processing (non-decision time (nDT) $\tau$).

The analysis revealed that only two parameters differed significantly across inequality contexts. In the advantageous context, perception was characterized by a higher weight on others ($\omega$) (ADV: 0.15±0.02, DIS: 0.10±0.02, ADV vs DIS: 95% CI [0.014, 0.10], Cohen's d = 0.44, $t(37)$ = 2.74, p = 0.037, Bonferroni correction), but the decision mechanism showed a lower decision threshold ($\alpha$) (ADV: 1.11±0.06, DIS: 1.34±0.09, ADV vs DIS: 95% CI [–0.33,–0.13], Cohen's d = –0.77, $t(37)$ = –4.78, p < 0.001, Bonferroni correction, *Figure 3A*). These findings indicate that when individuals are initially endowed with more money than the recipient, they require less evidence to take a choice, but crucially consider the others' payoffs more strongly.

Reassuringly, these differences in model parameters captured the key model-free behavioral differences between contexts: The individual probability to choose the equal option was significantly correlated with the weight on other-payoff (regression, ADV: $beta(35)$ = 0.07, SE = 0.005, p < 0.001; DIS: $beta(35)$ = - 0.04, SE = 0.005, p < 0.001) and the bias in starting point (DIS: $beta(35)$ = 0.01, SE = 0.005, p = 0.064; ADV: $beta(35)$ = 0.03, SE = 0.005, p < 0.001, *Figure 3B*). Similarly, individuals with a stronger bias in starting point towards the equal option took faster equal decisions and slower unequal decisions (Pearson correlation between the starting point and difference in response time between unequal choice and equal choice in DIS: $r(37)$ = 0.59, p < 0.001; and in ADV: $r(37)$ = 0.72, p < 0.001, *Figure 3C*). Irrespective of these differences between contexts, correlation analyses again suggested that people might employ a comparable overall decision mechanism across the two contexts, since all model parameters (*Figure 3—figure supplement 2*, minimum correlation: $r$ = 0.30, p = 0.06; maximum correlation: $r$ = 0.90, p < 0.001) and the probability to choose the equal option (*Figure 3—figure supplement 3*; $r$ = - 0.62, p < 0.001) were highly correlated across the two inequality contexts.

Together with the ERP data, these findings imply that even though participants took their choices in both contexts using a similar decision mechanism, various aspects of this mechanism (e.g. decision thresholds) and of the processes supplying the choice-relevant information (e.g. weights on other-payoff) differ across individuals and contexts. These behavioral and modeling findings motivated us to examine how exactly the *neural* processes extracting choice-relevant information from the stimuli may differ between the two inequality contexts.

## Contextual differences in altruistic choice reflect different neural processing of self-payoffs

To investigate neural processing of the choice-relevant information contained in the stimuli, we examined stimulus-locked ERP signals using general linear models (GLMs). Since the payoff options were presented sequentially, we assumed that people initially evaluate the changes (from the first to the second option) in their own or others' payoffs, and then integrate this information into the choice process. We thus switched from a choice-centered analysis approach (i.e. SSM and response-locked EEG analyses discussed above) to a stimulus-centered approach and analyzed how stimulus-locked ERPs reflected different aspects of the payoff information on the screen. To do so, we entered stimulus-locked ERP signals into a linear regression model (see Materials and methods for details) in which participants' own (partners') payoff change between the second and the first option [$\Delta S$ ($\Delta O$)] were included as predictors. This linear regression produced a set of estimated coefficients measuring

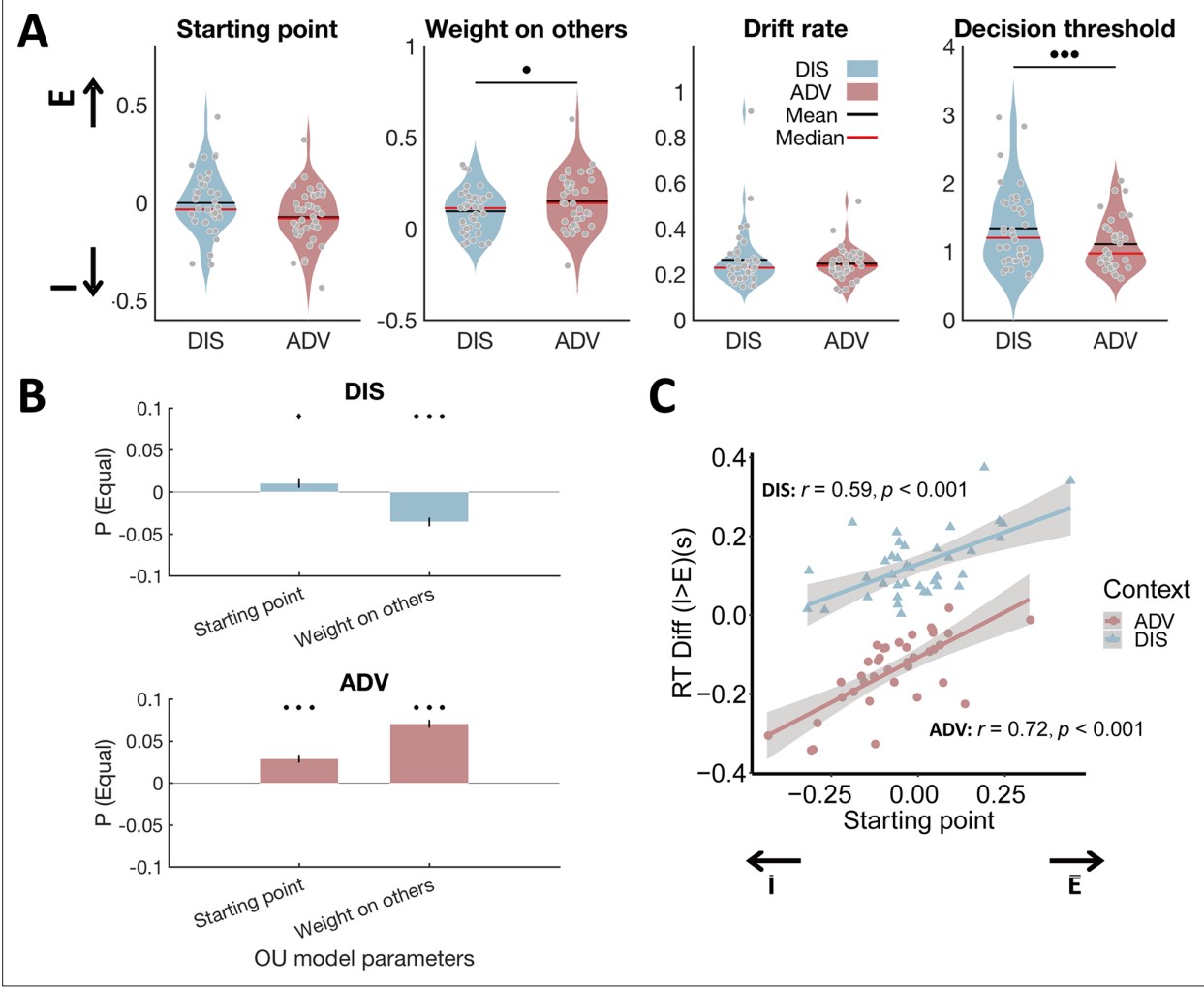

**Figure 3.** Model parameters differ between inequality contexts and show the expected relations to model-free behavioral data. (**A**) Relative to DIS, ADV increased weight on others and reduced decision threshold. Each grey dot represents one participant. E, starting point closer to equal option; I, starting point closer to unequal option. (**B**) Multiple regressions show that the probability to choose the equal option is related to a starting point closer to the equal option in both contexts, and a higher weight on others in ADV and a lower weight on others in the DIS context. Error bars indicate standard errors of the estimates, N = 38. (**C**) Individuals with starting points closer to the equal option responded more slowly to unequal relative to equal choices in both contexts. E, starting point closer to equal option; I, starting point closer to unequal option. •••, p < 0.001; ••, p < 0.01; ♦, p < 0.1.

The online version of this article includes the following figure supplement(s) for figure 3:

**Figure supplement 1.** OU model parameters in the two inequality contexts.

**Figure supplement 2.** Correlations of OU parameters between ADV and DIS contexts.

**Figure supplement 3.** Correlation of the probabilities to choose the more equal option in the ADV and DIS contexts.

**Figure supplement 4.** Model fits and parameters across inequality contexts.

**Figure supplement 5.** Correlations of parameters between the OU order model and the winning OU model (Inequality) in the DIS (**A**) and ADV (**B**) contexts.

the parametric effect strengths of $\Delta S$ and $\Delta O$ on neural activity for each channel and time window in each context and participant.

In these GLM analyses, we focused on the effect size measures and used permutation tests corrected for multiple comparisons (see Materials and methods for details) to quantify when and where EEG signals reflected self-payoff changes ($\Delta S$, $\beta 1$) and other-payoff changes ($\Delta O$, $\beta 2$). For the significant clusters emerging from these analyses, we then conducted post hoc analyses of raw ERP amplitudes to characterize the identified effects in more detail. This analysis approach allowed us to

test for two different possible scenarios of how context effects in distribution behavior may relate to neural perception processes.

On the one hand, while participants may always guide their choice by both the self-payoff change ($\Delta S$) and the other-payoff change ($\Delta O$), they may focus more strongly, or at different time points, on their own payoff or the receiver's payoff in the two contexts (Scenario 1) (*Harris et al., 2018*). On the other hand, participants may guide their choice more by fairness considerations (i.e. inequality context) that are in conflict with the motivations to maximize either self-payoffs or other-payoffs in the two contexts (*12, 14*). If this scenario holds, then ERPs should relate to self-/other-payoffs in opposite ways for the two contexts (Scenario 2). This is because a more unequal second option contains an increased self-payoff and/or decreased other-payoff in the ADV context, but the opposite pattern of a decreased self-payoff and/or increased other-payoff in the DIS context.

Our results support scenario 2: A conjunction analysis revealed no significant ERP component that showed similar effects of self-payoff change across the two contexts. Instead, when contrasting the whole-brain $\beta_1$ maps between the two contexts, we identified two different spatial-temporal clusters that showed opposite effects of self-payoff changes between contexts (i.e. earlier effect: from ~240 to 360ms after stimulus onset, $p_{cluster}$ = 0.04, *Figure 4A*; later effect: from ~440 to 800ms after stimulus onset, $p_{cluster}$ < 0.001, *Figure 4C*). In the earlier time window, more negative centroparietal neural responses were associated with self-payoff increases (positive $\Delta S$) in the ADV context (–0.09±0.04, Mean $\beta_1$ ± SE, 95% CI [-0.17,–0.01], Cohen's d = –0.36, $t$(37) = –2.23, p = 0.03) but with self-payoff decreases (negative $\Delta S$ in the DIS context (0.08±0.03, 95% CI [0.01, 0.15], Cohen's d = 0.40, $t$(37) = 2.45, p = 0.02; ADV vs. DIS: 95% CI [-0.28,–0.06], $t$(37) = –3.21, Cohen's d = –0.52, p = 0.003, *Figure 4A* right panel). In the later time window (~440–800ms after stimulus onset), however, more *positive* centroparietal neural responses related to self-payoff increases in the ADV context (0.10±0.03, 95% CI [0.04, 0.16], Cohen's d = 0.52, $t$(37) = 3.21, p = 0.003) but to self-payoff decreases in the DIS context (–0.12±0.03, 95% CI [-0.18,–0.05], Cohen's d = –0.60, $t$(37) = –3.68, p < 0.001; ADV vs. DIS: 95% CI [0.11, 0.32], $t$(37) = 4.30, Cohen's d = 0.70, p < 0.001, *Figure 4C* right panel). Thus, these ERP effects are more likely to be related to the processing of conflict between self-payoff change and inequality concern (scenario 2) rather than to common perceptions of self-payoff change across different contexts (scenario 1).

Interestingly, context effects in inequality-related neural processing were mainly evident in ERP correlations with self-payoff. The same analyses of $\beta_2$ maps revealed no ERP components related to other-payoff changes ($\Delta O$), neither across contexts nor differing between contexts. We also examined temporal dynamics of the parametric effect strengths of other-payoff change ($\Delta O$) in the specific clusters identified for the context-dependent effects of self-payoff change ($\Delta S$) in the above analyses but found no significant effects (*Figure 4—figure supplement 1*).

Post-hoc analyses of the ERP components corresponding to the effects identified in the above regression analyses also confirmed scenario 2. For this analysis, we categorized trials in each context into trials in which the second option increased self-payoff (i.e. more equal in DIS and more unequal in ADV) or decreased self-payoff (i.e. more unequal in DIS and more equal in ADV). For both time windows, ERPs associated with the increase and decrease of self-payoff yielded opposite neural effects in the two inequality contexts (see *Figure 4B and D*; for stats see Appendix 1). Importantly, a spatially unbiased interaction analysis between context (ADV vs DIS) and self-payoff change (Increased SP vs Decreased SP) in either time window revealed that people processed self-payoff-related inequality earlier in the advantageous context (*Figure 4D and E*, for statistical information, see Appendix 1 and Materials and methods; for a temporally agnostic ERP analysis confirming these results, see *Figure 4—figure supplement 2*). These timing differences thus suggest that behavioral differences in the two inequality contexts reflect differences in how and when the brain processes the differences in self-payoffs resulting from the choice options.

We also examined whether differences in altruism may relate to neural processing of the first option, by performing exploratory GLM analyses (similar to the above analyses) of the EEG signals elicited by the first option related to the size of the displayed self- and other-payoffs as well as their difference. In addition, we also examined whether EEG activity during the second option may potentially relate to the integrated decision value (i.e. utility difference between the chosen and unchosen option). However, these analyses did not reveal a significant cluster, as described in Appendix 1.

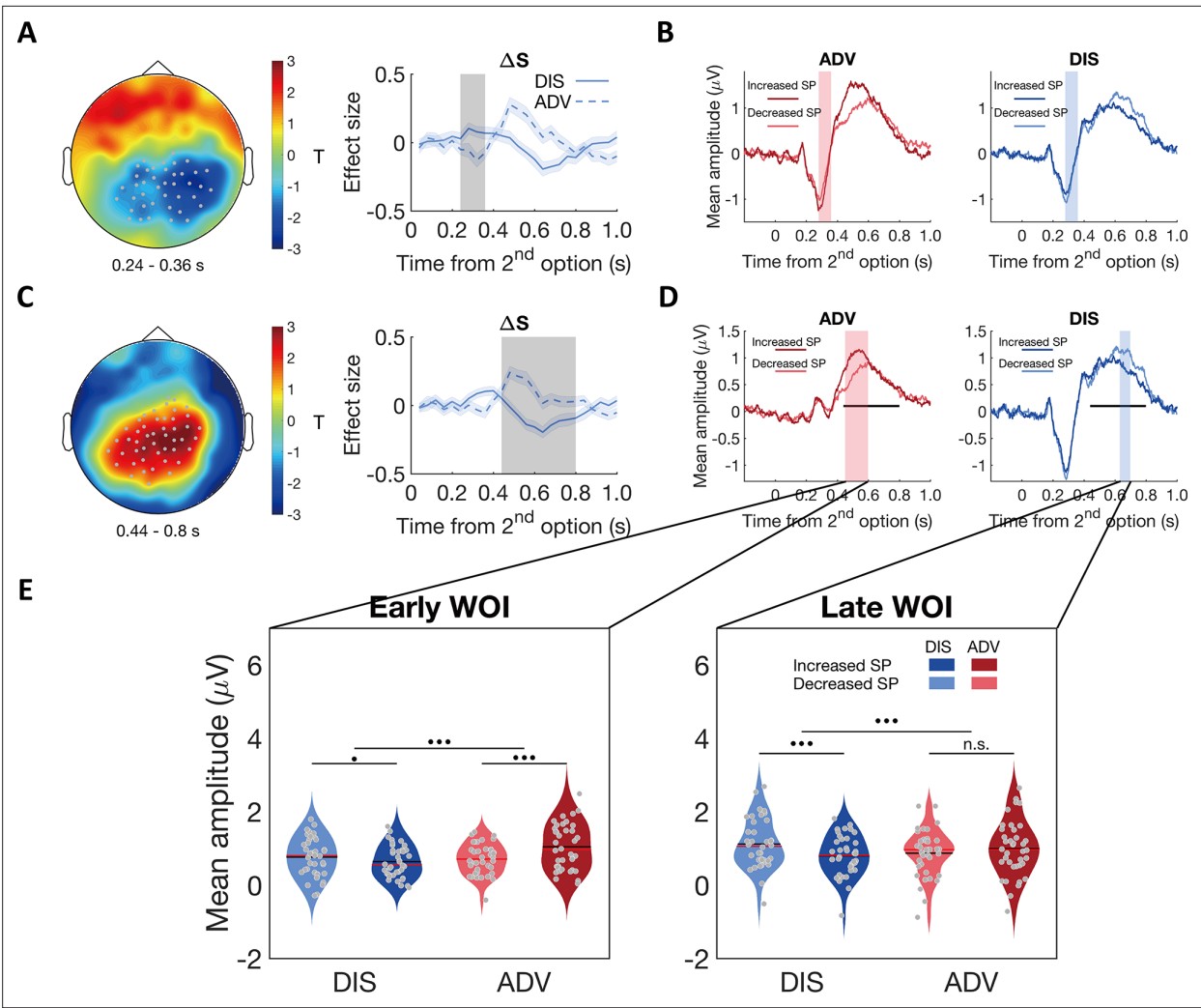

**Figure 4.** Contextual differences in neural processing of self-payoff change revealed by stimulus-locked ERPs. (**A**) & (**C**) ERP correlations with self-payoff change ($\Delta S$) show opposite signs for the two inequality contexts, in the time windows of ~240–360ms (**A**) and ~440–800ms (**C**): left panels, topographic scalp distributions of context differences; right panels, temporal dynamics of the parametric effect strengths of self-payoff change ($\Delta S$) in the identified clusters. Grey dots highlight channels that survived the threshold, grey shaded areas indicate the durations of the significant effects, and colored shaded areas indicate ±1 SEMs in (**A**) and (**C**), N = 38. (**B**) and (**D**) Average ERP waveforms for the parametric effect of $\Delta S$ in each context during the windows of ~240–360ms (**B**) and ~440–800ms (**D**) (For statistic information, see Appendix 1). Significant clusters surviving the cluster correction for multiple comparisons at p < 0.05 are reported. Increased SP, trials in which the second option increases self-payoff; Decreased SP, trials in which the second option decreases self-payoff. Colored shaded areas indicate the duration of significant interaction between context and self-payoff change in (**B**). Black lines indicate the duration of significant interaction between context and self-payoff change identified in (**C**), and colored shaded areas in (**D**) indicate separate durations of the significant interaction effect of self-payoff change and context over ERP waveforms in the time window identified in (**C**). (**E**) Self-payoff processing occurs at different time windows in the two contexts. For the significant cluster shown in (**C**), $\Delta S$ effects lasted from ~450 to 600ms after stimulus onset in ADV (**D**, pink shaded area), and from ~630 to 700ms after stimulus onset in DIS (**D**, blue shaded area). ERP responses are averaged magnitudes derived from each window of interest (WOI) in each condition. ERP responses to self-payoff change are stronger in an earlier WOI (~450–600ms, left panel) in the ADV context, and stronger in a later WOI (~630–700ms, right panel) in the DIS context. For the early WOI (left panel), increased self-payoff (Increased SP) was related to stronger neural responses than decreased self-payoff (Decreased SP) in ADV, but there was no difference in neural responses between decreased self-payoff and increased self-payoff in DIS (left panel). For the late WOI, decreased self-payoff (Decreased SP) was related to stronger neural responses than increased self-payoff (Increased SP) in DIS, but there was no difference in neural responses between increased self-payoff and decreased self-payoff in ADV (right panel). For statistical information, see Appendix 1. Increased SP, trials in which the second option increases self-payoff; Decreased SP, trials in which the second option decreases self-payoff. Red lines, Median; black lines, mean. •••, p < 0.001; •, p < 0.05; n.s., not significant.

The online version of this article includes the following figure supplement(s) for figure 4:

**Figure supplement 1.** Stimulus-locked ERP control analysis of other-interest by context.

**Figure supplement 2.** Whole brain analyses suggest that the $\Delta S$ effect on ERP responses occurs earlier in the ADV than in the DIS context.

## Individual differences in altruistic choice relate to differential neural processing of other-payoffs

Choice behavior did not just differ between the two inequality contexts but also across individuals. To investigate the neural processes underlying these individual differences, we divided participants into more altruistic (MA) and less altruistic (LA) based on their weight on others ($\omega$, median-split) and compared ERPs indexing neural processing of self-related and other-related payoffs (using the same GLM analyses and permutation tests corrected for multiple comparisons, see Materials and methods). Again, we directly investigated the two scenarios outlined above: Under scenario 1, behavioral differences between individuals may emerge from a (temporally) distinct focus on the processing of self- or other-related payoffs. By contrast, under scenario 2, the differences would be linked to differences in how inequality biases the processing of self- or other-related payoffs; these effects should reverse

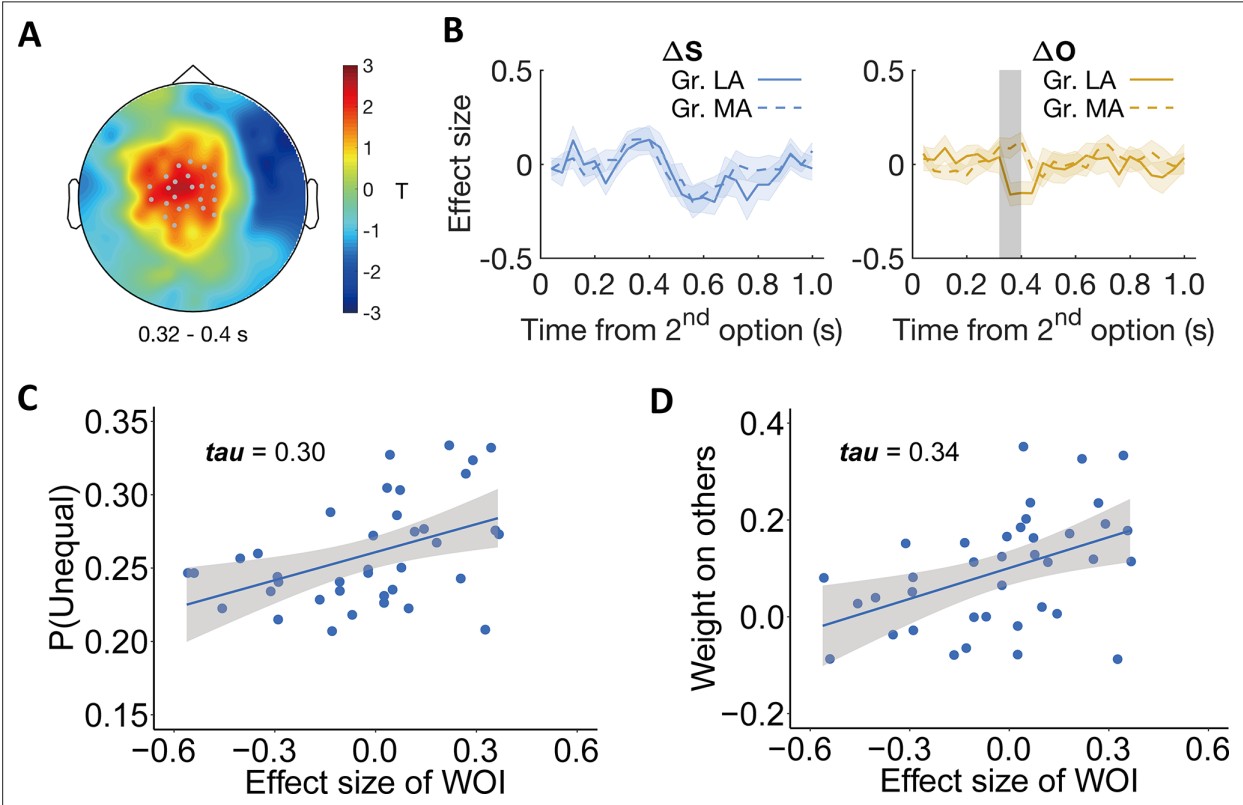

**Figure 5.** Individual differences relate to differential neural processing of other-payoffs in the stimulus-locked ERP analysis. Participants were divided into more altruistic (MA) and less altruistic groups (LA) based on the model parameter weight on others. (**A**) Topographic scalp distribution of the difference between MA and LA groups in ERP correlation with other-payoff difference ($\Delta O$) in the DIS context. The significant effect lasted from ~320 to 400ms after stimulus onset. Grey dots highlight channels that survived the threshold. (**B**) Specificity of the effect for $\Delta O$. Temporal dynamics of the parametric effect strengths of self-payoff change ($\Delta S$, left) and other-payoff change ($\Delta O$, right) in the identified cluster. Colored shaded areas indicate ±1 SEMs. Gr., group. Grey shaded area indicates the duration of the significant effect. Significant clusters surviving the cluster correction for multiple comparisons at p < 0.05 are reported, N(LA) = 19, N(MA) = 19. (**C–D**) Neural effects relate to behavior. The plots show the correlations between the effect strengths of other-payoff change ($\Delta O$) in the identified cluster and the probability to choose the unequal option (**C**) and the OU parameter of weight on others (**D**).

The online version of this article includes the following figure supplement(s) for figure 5:

**Figure supplement 1.** Correlation analysis between the model-parameter weight on others and neural responses to other-payoff change ($\Delta O$) in the DIS context.

**Figure supplement 2.** Stimulus-locked ERP analysis of other-interest by altruistic preferences.

**Figure supplement 3.** Individual difference analyses in the ADV context.

**Figure supplement 4.** Individual differences relate to differential neural processing of other-payoff in the stimulus-locked ERP analysis.

**Figure supplement 5.** Parametric effects of $\Delta S$ in the more altruistic (MA) and the less altruistic (LA) groups.

in the two contexts, since a more unequal second option implies an increased self-payoff and/or decreased other-payoff in ADV, but a decreased self-payoff and/or increased other-payoff in DIS.

The results of this analysis were more consistent with scenario 1, since individual differences were mainly related to variations in neural processing of other-payoffs, especially in the disadvantageous inequality context. Specifically, during DIS, other-payoff was associated with a more negative ERP response in the less altruistic group (LA: –0.16±0.06, Mean $\beta_2$ ± SE, 95% CI [-0.28,–0.04], Cohen's d = –0.64, $t$(18) = –2.78, p = 0.02; MA: 0.10±0.04, 95% CI [0.02, 0.19], Cohen's d = 0.56, $t$(18) = 2.46, p = 0.01). The effect was most strongly expressed over centrofrontal regions ($p_{cluster}$ = 0.03, post-hoc comparison: 95% CI [-0.43,–0.09], Cohen's d = –0.75, $t$(18) = - 3.27, p = 0.004, *Figure 5A and B*) and was also evident in a parametric correlation analysis between ERP responses to other-payoff and the weights on others' payoffs ($\omega$) (see *Figure 5—figure supplement 1*), but not for an analysis based on differences in starting point $\beta$ (*Figure 5—figure supplement 2*).

To visualize the relationship between this neural effect and both model-free and model-based measures of altruistic preferences, we extracted the individual effect size of other-payoff change ($\Delta O$) in the DIS context from the identified spatio-temporal cluster and correlated it with the individual probability to choose the more unequal option and with the weight on others ($\omega$) (to avoid circular inference, we only report and illustrate these correlations but do not compute corresponding p-values). This confirmed that the stronger the neural processing of the other's payoff difference (i.e., $\beta_2$), the more likely the participants were to choose the more unequal option (*Kendall's tau* (37) = 0.30, *Figure 5C*). The corresponding analyses with the model parameters also confirmed positive correlations with the individuals' weight on others (*Kendall's tau* (37) = 0.34, *Figure 5D*). The correlation analyses also confirm that these effects are only evident during the DIS context, since the corresponding correlations during the ADV context were minuscule (maximum tau-value = 0.007, minimum tau-value = –0.003, *Figure 5—figure supplement 3*). The same whole-brain analyses of individual differences were also performed in the ADV context, but no significant cluster was identified. For visualization of the specific origin of the neural effect identified in the individual-difference analysis of other-payoff processing, please see Appendix 1 and *Figure 5—figure supplement 4*.

To confirm that these individual differences in neural processing of choice-relevant information were unrelated to the context effects described further above, we also compared the more- versus less-altruistic groups in terms of how they processed self-payoff (as analyzed in the previous section). This showed that independent of the grouping parameter (i.e. weight on others or starting point), more altruistic and less altruistic groups exhibited comparable neural processing of self-payoff differences in those identified clusters (*Figure 5—figure supplement 5*). Thus, individual differences in altruistic preferences are linked to processing of other-payoff and not to temporal dynamics of the neural processing of self-payoff that was associated with contextual behavioral differences.

Taken together, these analyses suggest that although distribution choices are guided by a comparable neural decision process across different people and different contexts, there can be considerable contextual and individual differences in how the information fed into these processes is initially processed: While contextual differences in choice behavior mainly relate to differences in neural processing of self-related information, individual differences in altruism relate mainly to the processing of other-related information.

In addition, we also explored the neural correlates of individual differences by (1) correlating weight on others ($\omega$) with ERP signals related to the self-payoff change in each context, and (2) correlating differences in $\omega$ between contexts with the difference in ERP responses related to self-payoff change between contexts. However, these analyses did not reveal any significant cluster, as described in Appendix 1.

## Individual differences in altruistic preferences relate to synchronization of perception and evidence-accumulation signals

In previous studies, it has been observed that evidence accumulation processes need to integrate information from brain areas processing the stimulus dimensions relevant for choice and that this information exchange is characterized by inter-regional gamma-band coherence (*Polanía et al., 2015*; *Polanía et al., 2014*; *Siegel et al., 2008*). Given that our SSM-EEG analyses identified a comparable neural evidence accumulation process in different contexts, and that our ERP analyses revealed differential processing of the trial-wise information relevant for choice, we also examined how these two

sets of processes may exchange information to guide the decisions. Specifically, our finding that individuals' altruistic preferences (i.e. weight on others) in the disadvantageous context related to neural processing of other-payoff in centrofrontal signals led us to test whether the underlying brain areas exchange this information with the parietal areas accumulating this evidence for choice.

For each participant, we computed the debiased weighted phase lag index (dWPLI, *Vinck et al., 2011*) for gamma-band coherence between these two signals in the disadvantageous context, and compared this index of phase coupling between more- and less-altruistic participants (again median-split by $\omega$). We found that phase coupling between the centro-frontal regions (shown in *Figure 5A*) and the parietal regions (shown in *Figure 2C*) in the DIS context was indeed significantly higher for the more altruistic group (MA: 0.010 ± 0.003, 95% CI [0.004, 0.016], Cohen's d = 0.80, t(18) = 3.48, p = 0.003); LA: group ($7.75 \times 10^{-4} \pm 10 \times 10^{-4}$, 95% CI [–0.001, 0.003], Cohen's d = 0.17, t(18) = 0.76,

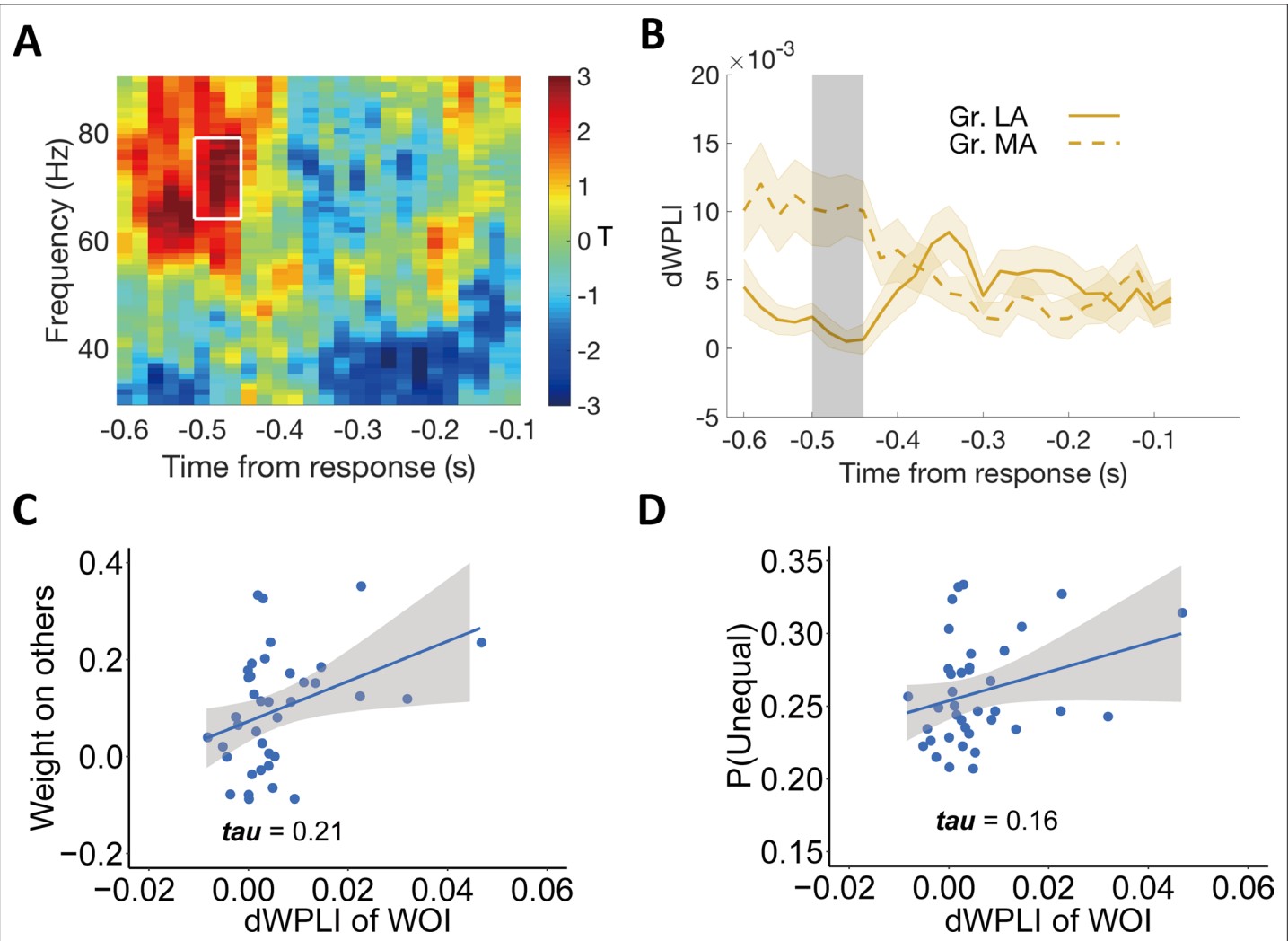

**Figure 6.** Individual differences in altruism relate to frontal-parietal synchronization. (**A**) Heatmap showing T-statistics for the differences between the more altruistic (MA) minus less altruistic (LA) group in phase coupling (dWPLI) between the frontal cluster shown in *Figure 5A* and the parietal cluster shown in *Figure 2C*, in the DIS context. A significant effect was identified in the gamma-band frequency range (~64–79 Hz) at the time window of ~520–460ms before response (highlighted in white box). (**B**) Temporal dynamics of the average dWPLI strengths in the ~64–79 Hz frequency range. Grey shaded area indicates the duration of the significant effect. Colored shaded areas indicate ±1 SEMs. Gr., group. Significant clusters surviving the cluster correction for multiple comparisons at p < 0.05 are reported, N(LA) = 19, N(MA) = 19. (**C**) Correlation between the strength of frontal-parietal synchronization (dWPLI) in the identified time-frequency cluster and the OU parameter of weight on others. (**D**) Correlation between the strength of frontal-parietal synchronization (dWPLI) in the identified time-frequency cluster and the probability to choose the unequal option in the DIS context.

The online version of this article includes the following figure supplement(s) for figure 6:

**Figure supplement 1.** Relationship between altruism and frontal-parietal synchronization in the ADV context.

p = 0.46, MA vs. LA: 95% CI [0.004, 0.015], Cohen's d = 0.99, $t$(18) = 3.38, p = 0.003). This effect was evident in the gamma band frequency (64–79 Hz) for the time window ~520–460ms before response ($p_{cluster}$ = 0.04, corrected for multiple comparisons, see Materials and methods and *Figure 6A and B*), and thus just before the onset of rise of the final evidence accumulation signals (occuring at around 400ms before response, cf with *Figure 2*). One may wonder to what extent the time window identified here (~520–460ms before response) temporally overlaps with the time window in which ERP responses to other-payoff differences are associated with individuals' decision weight on others (~320–400ms after stimulus onset). To test the overlap of these two time windows, we computed for each trial the temporal distances of the start and end points of the time window of ~320–400ms from the response and defined these distances as the transferred response-locked time points for each trial. When averaging these transferred response-locked time points across trials for each participant, we found that the response-locked time window for the period ~320–400ms after stimulus onset ranged from ~530 (95% CI [600 – 470]) to 450 (95% CI [520 – 390]) ms before response and therefore contained the time window in which we found increased phase coupling (~520–460ms before response). This confirms that the time window in which we find increased gamma-band coherence indeed overlaps with the time window in which ERP responses to other-payoff differences are associated with the individual decision weight on others.

To visualize the hypothesized relationship between this neural effect and both model-based altruistic preferences and model-free unequal choice, we extracted the individual effect size in the DIS context from this temporal-frequency cluster and correlated it across participants with the weight on others ($\omega$) and with the individual probability to choose the unequal option (note again that these analyses are for illustrative purposes and do not constitute circular inference). As expected, the strength of gamma-band synchronization was related to the weight on the others' payoff (*Kendall's tau* (37) = 0.21, *Figure 6C*) and to the frequency of choosing the unequal option in the disadvantageous context (*Kendall's tau* (37) = 0.16, *Figure 6D*).

The corresponding analyses for the advantageous context did not reveal any significant temporal-frequency cluster (see *Figure 6—figure supplement 1*), which is in line with the findings that in this context, there was also no relation between individuals' altruistic preferences (i.e. weight on others) and ERP responses (see above). Taken together, these results suggest that frontal-parietal gamma synchrony may serve to incorporate information about other-payoff in evidence accumulation processes to promote altruistic preferences under disadvantageous inequality. Moreover, they suggest that while distribution decisions in different contexts and individuals are governed by comparable neural decision processes, individual differences in altruism may not only originate from differences in the perception processes but also from how this payoff-related information is shared between different regions by neural coherence.

One may wonder whether the frontal-parietal dWPLI functions as a mediator to account for the effect of ERP responses to the other-payoff change on individuals' altruistic preferences ($\omega$). However, our analyses suggested that the mediation model could not be established. Detailed descriptions of the analyses and results are reported in Appendix 1.

## Discussion

The psychological and neural mechanisms underlying altruism are debated intensely. On the one hand, evolutionary theories suggest that relatively simple, unified mechanisms may have evolved to facilitate actions that run counter to our self-interest to benefit others (*Gintis et al., 2003*; *Piliavin and Charng, 1990*). On the other hand, models from economics and psychology have suggested that altruism is complex and may draw on a whole range of different motives that can be differentially triggered in different contexts and different individuals (*Bester and Güth, 1998*; *Côté et al., 2015*; *Hein et al., 2016*). Here we investigated – using computational modeling and EEG methods with high temporal resolution – what cognitive processes that temporally evolve during the decision drive contextual and individual differences in altruism. This allowed us to investigate whether these differences reflect the differential use of integrative valuation and decision processes throughout later stages of the decision, or whether they reflect biases in the initial processing of the choice-relevant information before this information is integrated into a common evidence accumulation process leading up to the final choice.

Our findings are more compatible with the latter: We find that a comparable response-locked parietal evidence accumulation process is deployed across different contexts and in different individuals, but also considerable differences in how the choice-relevant information is initially processed. Context differences were most evident in neural signals related to self-payoff processing, whereas differences between individuals were most apparent in varying neural processing of other-payoff. Specifically, the sequential sampling model and response-locked EEG results suggested that the final evidence accumulation process in the two contexts recruited the same computations and the same brain areas. In contrast, the early processing of choice-relevant information (indexed by stimulus-locked ERPs) differed in terms of both recruited computations (people focusing on different aspects of information) and neural implementation, as EEG signals were observed in different brain areas or at different time windows in different contexts or individuals. Thus, contextual and individual differences in altruism do not reflect the use of different integrative decision processes, but are linked to differences in how the choice-relevant information is initially perceived and/or attended to before being integrated into evidence-accumulation mechanisms.

## Neural sources of variability in altruism

Our study provides critical insights into the nature of altruism. Since altruism can co-occur with different types of motivations under different circumstances (*Hein et al., 2016*; *Vekaria et al., 2017*), it is crucial for us to pin down whether these motivations are expressed as fundamental differences in how choices are taken, or as variations in initial (perceptual/attentional) information processing. Either of these possibilities would have different implications for our understanding of the nature of human altruism, why people differ in their altruistic tendencies, and how these tendencies may be altered by interventions (*Gao et al., 2018*; *Güroğlu et al., 2014*; *Jiang et al., 2016*; *Krajbich, 2018*; *Teoh et al., 2020*; *Tricomi et al., 2010*).

Previous neuroimaging studies have been ambiguous on this point, showing that both common and distinct brain networks are active during altruistic decision making in different inequality contexts (*Gao et al., 2018*; *Morishima et al., 2012*; *Yu et al., 2014*). However, it is largely unclear what specific neuro-cognitive mechanisms underlie these different patterns of brain activation. Our findings fill this gap, by revealing that altruistic choices in different contexts are guided by a comparable neural decision mechanism in similar brain areas that is well described by a sequential sampling model; the activity patterns we observed are similar to neural mechanisms reported to be recruited for perceptual and non-social value-based decision making (*Pisauro et al., 2017*; *Polanía et al., 2014*). Thus, our results support the view that altruism is not a functionally distinct aspect of human behavior but rather controlled by processes that are tightly integrated with general control processes required for non-social aspects of our actions (*Krajbich et al., 2015*).

Contextual and individual differences in altruism appear to be related to perceptual and or attentional processing of the choice-relevant information before this information is integrated into the evidence accumulation process. It has been long established in the decision making literature that attention to specific decision-relevant information can determine individuals' final choices (*Ghaffari and Fiedler, 2018*; *Krajbich, 2018*; *Krajbich et al., 2010*) and even shape social preferences (*Jiang et al., 2016*; *Teoh et al., 2020*). For example, higher time pressure will bias gaze toward stimuli relevant to one's interests, and the amount of this bias can predict changes in altruistic preferences across individuals (*Teoh et al., 2020*). Our neural results suggest perceptual or attentional origins for contextual and individual differences in altruism even under normal situations without time pressure. These differences are linked not just to perceptual processing of self- or other-payoffs (*Hutcherson et al., 2015*) but also to the neural processing of inequality information in different contexts (*Yu et al., 2014*). This extends previous reports that attention can affect processing of self-interest (*Teoh et al., 2020*) to neural processing of others' interests and to how this information is communicated between different brain regions.

## Parietal evidence accumulation extends to social decisions

Our findings also enhance our understanding of the generality of parietal evidence accumulation signals for different aspects of human decision making. The involvement of parietal regions in evidence accumulation processes was initially established by perceptual decision making studies with electrophysiology and EEG signals (e.g. direction discrimination task), in which the build-up rate of neural

activity in parietal regions was considered as a model-free measure of EA process (*Brosnan et al., 2020*; *Churchland et al., 2008*; *Kelly and O'Connell, 2013*; *Kiani and Shadlen, 2009*; *O'Connell et al., 2012*; *Shadlen and Newsome, 2001*). Recently, a similar role of parietal regions in evidence accumulation during value-based decision making was proposed by EEG studies of food choices based on subjective preferences (*Pisauro et al., 2017*; *Polanía et al., 2014*). One may wonder whether these neural signals may reflect low-level motor planning, rather than high-level decision mechanisms linking decision values to motor responses. However, we believe this is not plausible, for two reasons. First, a recent simultaneous EEG-fMRI study has suggested that the parietal evidence accumulation EEG signals can be traced to activity in the posterior medial frontal cortex, which functions with value processing regions (i.e. ventromedial prefrontal cortex and striatum) to link decision values to motor responses, rather than reflecting pure motor preparation (*Pisauro et al., 2017*). Second, since basic motor planning signals are typically observed in the hemisphere contralateral to the response hand (*Kornhuber and Deecke, 2016*; *Schurger et al., 2021*), the action readiness potential should be distributed more widely in the left hemisphere (all participants responded with the right hand). Such signals related to motor planning should thus exhibit fundamentally different spatial-temporal characteristics than the parietal evidence accumulation signals observed here. Moreover, the causal role of parietal regions in evidence accumulation underlying decision making has been established in studies with both animals and humans (*Licata et al., 2017*; *Polanía et al., 2015*; *Yao et al., 2020*; *Zhong et al., 2019*; *Zhou and Freedman, 2019*). Inactivation of projections from parietal regions to sensory-specific regions (e.g. auditory cortex) will not only prolong rats' evidence accumulation processes but also impair their decision performance (*Yao et al., 2020*; *Zhong et al., 2019*). Consistent with this, disrupting synchronization between parietal and frontal regions was also found to causally decrease the precision of value-based decisions (*Polanía et al., 2015*).

In the current study, we not only revealed comparable evidence accumulation signals in the parietal region in both inequality contexts but also validated the tight relation of the model-free ERP build-up rate with the SSM model-based parameter (i.e. drift rate) for altruistic decision making. This correlation relates the parietal EEG signals to the decision mechanism captured by the SSM, and also confirms that the observed response-locked EEG signals are involved in linking decision values to responses, rather than just in the pure motor planning (*Kelly et al., 2021*; *Pisauro et al., 2017*; *Polanía et al., 2014*; *Schurger et al., 2021*). Since both SSM model-based neural EA signals and model-free CPP signals reflect relatively late neural processes that are closer to the ultimate responses than the initial perceptual processing, our results do not exclude the possibility that different systems are involved in the perceptual processing of the choice-relevant information at earlier stages. In fact, our findings provide evidence that although there may be comparable final decision processes guiding behavior in different contexts, these processes may receive inputs from different earlier neural processing of stimulus, and that these mechanisms may differ in different contexts. Thus, it appears that even high-level social decision making involves parietal evidence accumulation signals, rendering these signals a close approximation of a unified decision mechanism that can flexibly draw on many different types of information to guide choice (*Kumano et al., 2016*). Future studies using brain stimulation techniques may be needed to establish the causal role of parietal regions in evidence accumulation of social decision making.

## Inequality processing is crucial for altruistic decisions in disadvantageous and advantageous contexts

Although it is often observed that attention to specific aspects of decision-relevant information can bias individuals' decisions (*Jiang et al., 2016*; *Krajbich et al., 2010*; *Maier et al., 2020*; *Teoh et al., 2020*), it is still unclear whether these biases reflect enhancement of the behavioral impact of single dimensions of choice-relevant information (i.e. here self-payoff or other-payoff), or of higher-level integration of various stimulus aspects (i.e. here integration of self- and other-payoff with inequality contextual information). Both these possibilities have been incorporated in computational models of the evidence accumulation process, with little decisive evidence to back up either option.

Consistent with behavioral and modeling results, our EEG results suggest that altruistic decision making in both inequality contexts involves integrated processing of self-payoff with inequality information, rather than separate modulation of processing for self-payoffs, other-payoffs, and inequality information. The self-payoff difference between the two options evoked ERP components in the early

time window (~240–360ms after stimulus onset) and a late time window (~440–800ms after stimulus onset). Both these effects showed different temporal dynamics in different inequality contexts, and they occured at similar time windows as typical ERP components related to anticipation, cognitive control, and attentional processing (e.g. N2 and FRN lasting ~200–300ms, P300 peaking at ~300–400ms, or P600 lasting occurring ~400–800ms post-stimulus onset *Coulson et al., 1998*; *Sutton et al., 1965*). Our finding that inequality context fundamentally alters neural processing of stimulus information during both these time windows suggests that inequality is a fundamental motivational component that impacts the brain signal already during the earliest stages of perceptual processing and that guides subsequent attentional processing to bias choice.

Recent value-based decision making research also showed a late positivity response in a similar time window, varying with the value of stimulus attributes relevant to decision making (*Harris et al., 2018*). These studies consistently suggested that a stronger late positivity is evoked when stronger attention or cognitive resources are required to process stimuli with higher complexity or saliency. In our study, the effects in the late positivity responses imply that this component does not necessarily encode the value of self-payoff, but may participate in the integrated processing of self-payoff with inequality information, with different temporal dynamics across the two contexts. This temporal difference in neural signals may also account for the observation of shorter reaction times in the ADV compared to the DIS context, with faster conflict resolution associated with shorter response times (*Chen and Fischbacher, 2020*). Another potential (but more speculative) explanation for these findings could be people may differentially re-orient attention to different option attributes. That is, people may generally attend first to their own payoffs, but more altruistic people may then be more likely to attend others' payoffs (*Teoh et al., 2020*), so that participants may be differentially taxed in different contexts by the 'meta-decision' of where to direct attention to guide their decisions. Although we ruled out eye movements in the current study, participants in the advantageous context may process their own payoffs faster and re-orient attention to others' payoffs earlier than in the disadvantageous context, where they may need to take more time to process their (lower) own payoffs. Since we did not manipulate or measure attention orientation in the current study, we cannot arbitrate between these different explanations for the observed effects, which may perhaps be further investigated by eye-tracking studies. Irrespective of these possible explanations, our findings again underline that the processes involved in controlling altruism in different inequality contexts differ mainly in terms of how the decision-relevant information guiding choice is perceived and attended to (*Gao et al., 2018*; *Güroğlu et al., 2014*; *Yu et al., 2014*).

Note that the sequential presentation of options in the current study may have made it difficult to examine when and how the human brain evaluates raw values of self- and other-payoffs, instead of the payoff differences. Moreover, the low spatial resolution of EEG may also make it difficult to detect neural signals related to self-/other-payoff evaluations implemented in subcortical regions. Therefore, our approach may not provide a full picture of all neural computations underlying such decisions. Nevertheless, our results provide new insights by clarifying the temporal characteristics of altruistic choices across different cognitive processing stages (i.e. early perceptual/attentional vs. late decision processes).

## Individual differences are linked to the distributed processing of others' interests

A core question in research on altruism has always been why people differ in their altruistic motivation and behavior (*Gao et al., 2018*; *Morishima et al., 2012*; *Yu et al., 2014*). Our EEG results add a novel perspective to this literature, by showing that individual differences in altruistic preferences (i.e. the decision weight placed on others' payoffs) are related to early brain processing and fronto-parietal information exchange of the information quantifying other-interest. While this highlights that perceptual and attentional processing of choice-relevant information may strongly influence choice outcomes even for high-level social choices, it is noteworthy that individual differences in altruism were linked to the processing of different aspects of the choice situation (others' outcomes) than those aspects that seemed related to contextual differences (inequality context). This suggests that different perceptual/attentional biases underlie variations in altruism across situations and people, even though these variations may at first glance appear to be of similar magnitude. Thus, our results demonstrate more generally that neural measures can help us to better understand the fundamental

motivations underlying variability in behavior across people and contexts, which would be impossible to achieve with behavioral measures alone (*Hein et al., 2016*).

Our finding that frontoparietal gamma-band coherence correlated with altruism identifies a specific neural mechanism that may underlie how the information of other-interest is integrated into choices. This process shows clear parallels with previous findings from perceptual and non-social value-based choices, where related increases in gamma-frequency band coherence have also been shown to facilitate information transfer between distant brain regions to improve performance (i.e. precision) (*Gregoriou et al., 2009*; *Polanía et al., 2014*; *Vinck et al., 2013*; *Womelsdorf et al., 2006*). Specifically, frontal regions involved in assigning value to choice options have been found to functionally couple with parietal regions in the gamma band to implement evidence accumulation in value-based social decision making (*Basten et al., 2010*; *Philiastides et al., 2010*; *Pisauro et al., 2017*), and one study even demonstrated that stronger frontal-parietal phase coupling in gamma band can causally increase the precision of value-based decision making (*Polanía et al., 2015*). Based on these earlier findings and our present results, it appears possible that stronger information transfer or sharing of others' payoff between frontal and parietal regions can indeed account for greater altruistic preferences in social decision making. Nevertheless, such modulation of frontal-parietal interregional information transfer on altruistic preferences may be relayed by subcortical regions, as suggested by a recent monkey study showing that stronger gamma-band synchronization between anterior cingulate cortex (ACC) and basolateral amygdala is related to enhanced other-regarding preferences (*Dal Monte et al., 2020*). Given the limitation of spatial resolution in EEG recordings, we cannot pinpoint the involvement of specific cortical or subcortical regions, but regions like ACC and amygdala could be candidate regions involved in information transfer underlying altruistic preferences in humans as well.

Our findings that SSM model parameters and response-locked ERPs were very similar across the two contexts support the hypothesis that participants make decisions in both contexts using a similar final integrative decision mechanism that is well captured by a sequential sampling model. By contrast,the stimulus-locked ERP signals exhibited categorically different patterns between the two contexts. This statistical uncoupling shows clearly that these two types of signals (evidence accumulation signals and response-locked ERPs on the one hand vs stimulus-locked ERPs on the other) index fundamentally different aspects of the neurocognitive processes unfolding during decision making: The former relate to the signals integrating various sources of information that relevant for a decision, whereas stimulus-locked ERP signals are more temporally close to early perceptual, attentional, or valuation processing of each source of choice-relevant information. Contextual differences in these signals thus reflect how the neural processes extracting choice-relevant information from the stimuli differ between the two inequality contexts. Therefore, our findings suggest that while the computation/implementation of early processing of choice-relevant information differs, the computation/implementation of the late evidence accumulation process is similar across contexts and can thus be viewed as a unitary decision mechanism that conceptually links these two types of choice contexts.

It is also noteworthy that our findings at first glance appear not fully consistent with previous MRI studies suggesting that TPJ may be a critical region for individuals' altruistic preferences in only the ADV context (*Morishima et al., 2012*). Specifically, we did not observe any significant neural responses encoding other-payoff changes or any significant correlations between EEG signals and individuals' weight on others in the ADV context. It is difficult to clarify whether the discrepancy between our results and previous fMRI findings really resulted from inconsistent neural activity or from the insensitivity of EEG in detecting signals from the specific brain region recruited in the previous fMRI study. Future studies that simultaneously collect EEG-fMRI data may help to address this issue. Instead, we observed modulation of local and large-scale processing of other-payoff change in the DIS context in which individuals' weights on others' payoffs are lower than that in the ADV context. One potential explanation for this contextual difference is that individuals' relatively disadvantageous position in the DIS context may make selfish people more inclined to upward social comparison (i.e. compare themselves with those who are better off), and thus may make them more sensitive to the profit changes of others who already possess more than themselves (*Boyce et al., 2010*; *Payne et al., 2017*). In contrast, the ADV context may bias participants to more uniformly consider others' profit – as suggested by our modeling results and previous studies (*Morishima et al., 2012*) – thereby

potentially reducing individual differences in sensitivity to others' profit change. Nevertheless, future studies are needed to confirm this speculation.

Our study also suggests that the effects of inequality contexts on the decision process may relate to framing effects that are tightly related to how information is presented to participants (*Dietze and Craig, 2021*). In the current paradigm, the two allocation options were presented sequentially, so that participants knew the context they were in before the EA process started with the presentation of the second option. This may explain why the inequality contexts affected altruistic decisions not just by modulating perceptual processing, but also by changing other latent decision processes such as the criterion required for a decision to be taken (i.e. decision threshold). The higher payoffs in the ADV context may put participants in a relative gain frame and render them more impulsive (i.e. lower decision threshold), since they benefit more than the others from both options, whereas the lower payoffs in the DIS context may put participants in a relative loss frame and drive them to be more cautious (i.e. higher decision threshold) and integrate more evidence to make the final decision (*Diederich and Busemeyer, 2006*; *Dietze and Craig, 2021*). Such contextual differences in decision threshold can also explain the contextual difference in response times, with a lower decision threshold leading to faster response times in the ADV compared to the DIS context.

Importantly, our experimental setup is similar to situations in real life, since people usually know whether they have more or less money than others when deciding whether to behave altruistically. Nevertheless, our results emphasize that future studies should examine how different information presentation formats may bias different latent decision processes, such as attention to different aspects of information (e.g. self-payoff, other-payoff, or inequality level) and impulsivity, to influence social preferences and decisions, which has rarely been investigated in previous studies (*Gao et al., 2018*; *Hutcherson et al., 2015*; *Morishima et al., 2012*).

## Facilitation of altruistic behavior

Our findings provide empirical evidence for improving strategies to promote altruism from the perspective of decision processes. Most previous studies focused on clarifying the relationship between individual variations in altruistic preferences and dispositional empathy (i.e. empathy-altruism hypothesis) and suggested that facilitating individuals' empathic concerns for others will enhance altruistic preferences or behavior (*Batson et al., 2007*; *FeldmanHall et al., 2015*; *Hein et al., 2010*; *Lockwood et al., 2017*). However, it is possible that corresponding interventions meant to enhance empathy do not operate mainly by enhancing emotional concern for others but rather by shifting attention to corresponding information. This would fit our findings that variations in altruism across different contexts and individuals may have more perceptual or attentional origins and suggests that it may be possible to have substantial influences on altruistic behavior by interventions that shift attention to related choice-relevant information even without explicitly cuing or training empathic concern (*Jiang et al., 2016*; *Teoh et al., 2020*).

Our approach to examining altruism across contexts can also help to improve diagnosis and treatment for socially apathetic people or other people with a high subclinical or clinical level of psychiatric and neurological disorder (e.g. psychopathy, autism, or alexithymia), by investigating which latent components of decision processes underlying altruistic behaviors may be altered in these disorders and changed by corresponding treatments (*Bird and Viding, 2014*; *Feldmanhall et al., 2013*).

Taken together, the SSM-EEG framework we employed here provides a powerful tool to clarify whether changes in altruistic preferences across contexts and individuals reflect fundamentally different motives and use of different decision processes, or rather different perceptual/attentional biases in the processing of payoff information that may be amenable by corresponding interventions. More generally, our study suggests that this approach may be employed to study neural origins of individual and contextual variations also in other types of social decision making that need to integrate value signals from multiple sources, such as empathy-based or reciprocity-based altruism, moral decisions, and sophisticated strategic decision making.

# Materials and methods

## Participants

Forty-one participants (16 females) participated in the study. Participants were informed about all aspects of the experiment and gave written informed consent. All the participants were right-handed, had normal or corrected-to-normal vision, were free of neurological or psychological disorders, and did not take medication during the time the experiment was conducted. They received between 75 – 122CHF (depending on the realized choices) for their participation. Three participants were excluded from analyses due to excessive EEG artifacts (the remaining 38 participants: 19–31 years of age, mean 24.5 years). The sample size was determined based on a previous study (*Hutcherson et al., 2015*) that detected a significant DDM parameter of weight on others with an effect size of d = 0.4. We thus determined our sample size based on d = 0.4 with G*Power 3.1, which suggested that we need 41 participants to have adequate power ($1 – \beta > 0.80$) to detect an effect with d = 0.4 at the level of $\alpha$ = 0.05. The experiment conformed to the Declaration of Helsinki, and the protocol was approved by the Ethics Committee of the Canton of Zurich (KEK: 2011-0239/3).

## Task

Each participant made 416 real decisions in a modified Dictator Game, requiring them to choose one of two allocations of monetary amounts between the participant and a receiver (another participant in the same study). We used two inequality contexts: In the disadvantageous context (DIS), the token amounts for participants were always lower than for the receiver; in the advantageous context (ADV), they were always higher. On each trial, one of the two options was revealed at the beginning of each trial (i.e. the reference option), and the other option was revealed at a later time (i.e. the alternative option, see below and *Figure 1A* for details). There were four levels of reference options (i.e., 24/74, 24/98, 46/74, and 46/98) in each context. For the DIS context, the numbers before the slash denote the token amounts allocated to the participant, and the numbers after the slash denote the token amounts given to the partner and vice versa for the ADV context. There were 26 alternative option levels corresponding to each reference option level, with half of them being more equal than the reference option and the other half being more unequal than the reference option. The token differences for each party between the alternative and the reference options ranged from –19 to 19. To make sure that our trial set can capture a large range of altruistic preferences and can properly estimate the parameters of the Charness-Rabin model, we not only included decisions requiring opposite changes in self- and other-payoff (i.e. maximizing one and minimizing the other), but also a small number of choice options leading to changes in inequality but increases or decreases of both self- and other-payoffs. To avoid repetitions of exactly the same choices, we included two different trial sets with the same reference options and similar distributions of alternative options. The second trial set was generated by adding a random jitter (i.e. - 1,+0, or + 1) to self-/other-payoffs of the alternative options of the first trial set (*Figure 1—figure supplement 1*).

At the beginning of each trial, participants saw a central fixation cross together with a reference allocation option on one side of the cross. The alternative option, to be shown on the opposite side of the fixation cross, was initially hidden and replaced with 'XX' symbols. Participants were asked to keep their eyes fixated on the central cross for at least 1 s (this was controlled by the use of eye tracking, see below). Only after successful fixation for at least 1 s, the font color for all the stimuli changed from blue to green (the change direction of font color was counterbalanced across participants), indicating the initiation of the trial. After a temporal jitter of 0.5–2 s (a uniform distribution with a mean of 1.25 s), the alternative option was revealed (with XX symbols replaced with actual amounts). Participants had to choose the left or right option by pressing the corresponding keys on the response box with the right index or middle finger within 3 s. The selected option was highlighted with a color change. Note that since the fixation was constrained during the whole trial before participants' response (for a more detailed description, see the section on Eye tracking), the retinal processing of the first and the second option should be the same. Since all payoff stimuli were presented close to the fixation cross (i.e. visual angle smaller than 3°), participants were able to see the numbers clearly without shifting their gaze.

The task was divided into 4 blocks, with each block lasting around 10 min. In total, the experiment took around 50 min to 1 hr, including breaks between blocks.

To avoid potential experimental demand effects, we did not explicitly describe the classification of decision contexts as advantageous or disadvantageous in the instructions. Nevertheless, since participants were presented with the first option at the beginning of each trial, they were aware of whether they were better or worse off than the other at the beginning of each trial. Such a sequential presentation resembles real-life contexts in which people are usually already aware of whether they are in a better (advantageous) or worse (disadvantageous) position, before having to decide whether to improve the welfare of the other.

To avoid potential attentional or visual processing bias due to fixed position of reference/alternative payoffs, we counterbalanced stimulus positions (left versus right) trial-by-trial within the participant. To lower the processing load and avoid potential response errors due to misrecognition, we fixed the position of self/other payoffs within-participant and counterbalanced it between participants.

The participation payment was determined at the end of the experiment and consisted of three parts: a base payment of 45 Swiss Francs (CHF, around $45 at the time of the experiment), one bonus (Bonus A) payment that depended on participants' own decisions, and another bonus (Bonus B) payment that was determined based on the choices of previous participants whose outcomes were allocated to the current participant. To determine these bonus payments, the participant drew two envelopes from two piles of envelopes, one pile for bonus A and one pile for bonus B. Each envelope contained five different randomly determined trial numbers.

For Bonus A, the numbers in the envelope were the trial numbers that would be selected from the full list of the participant's choices to be paid out. We calculated the mean payoff from these options and paid this as the first bonus.

For Bonus B, the numbers in the envelope were chosen from the full list of choices taken by previous participants (including behavioral pilot studies). This list of choices was randomly drawn from the full list of choices of all previous participants so that the participant was randomly paired with a different person on every round. The mean of the partner's payoffs across the chosen five rounds was paid out as the second bonus.

The partner's payoffs resulting from the participant's own choices also entered the full list of choices for future participants, meaning that they were to be paid out to these participants if any of the current participant's choices were selected. The exchange rate was 1 token = 0.5 CHF.

## EEG recordings

EEG signals were recorded from 128 scalp sites using sintered Ag/AgCl electrodes mounted with an equidistant hexagonal layout using a Waveguard Duke 128 channels cap (http://www.ant-neuro.com/). The cap was connected to a 128-channel QuickAmp system (Brain Products, Munich, Germany). All EEGs were referenced online to an average reference electrode. Electrode impedance was kept below 5 kΩ for all the electrodes throughout the experiment. The bio-signals were amplified with a bandpass from 0.016 to 100 Hz and digitized online with a sampling frequency of 500 Hz. The EEG cap was set up on each participant's head before he/she entered the soundproof and electromagnetically shielded chamber to perform the decision-making task during EEG recordings.

## Eye tracking

We used an EyeLink-1000 (http://www.sr-research.com/) to track and record participants' fixation patterns with a sampling frequency of 500 Hz. Before each trial, participants were instructed not to blink and keep their eyes fixated at the central fixation cross for 1 s before the trial started. From the start of the trial until the response was detected, participants were also asked not to blink and maintain fixation (tolerance 3°). If they failed to do so, the trial was aborted, and participants were informed with a message reminding them that the trial was invalid due to eye movements. Such trials were discarded in the EEG analyses. Participants practiced the task and fixation in a 10 min practice session. For this practice session, participants were presented with a different set of allocation options than those used in the real experimental session.

## Computational model

We used the Ornstein-Uhlenbeck (OU) process to model evidence accumulation underlying individuals' decisions. The OU process updates the evidence (EA) at each subsequent time step $t$ with the following equation:

$$EA_{(t+1)} = EA_{(t)} + \left( \lambda_{(c,\,s)} \times EA_{(t)} + \kappa_{(c,\,s)} \times VD_{(c,\,s,i)} \right) dt + N\left(0, \sigma\right)$$

$$\mathrm{VD}_{(c,\,s,i)} = \left( \left(1 - \omega_{(c,\,s)}\right) \times \mathrm{E}^{S}_{(c,\,s,i)} + \omega_{(c,\,s)} \times \mathrm{E}^{O}_{(c,\,s,i)} \right) - \left( \left(1 - \omega_{(c,\,s)}\right) \times \mathrm{I}^{S}_{(c,\,s,i)} + \omega_{(c,\,s)} \times \mathrm{I}^{O}_{(c,\,s,i)} \right)$$

with indices c for conditions (c = DIS for disadvantageous inequality context, c = ADV for advantageous inequality context), s for participants (s = 1,..., $N_{participants}$), and i for trials (i = 1,..., $N_{trials}$). $\mathrm{E}^{S}_{(c,\,s,i)}$ $\left( \mathrm{I}^{S}_{(c,\,s,i)} \right)$ indicates participants' payoff of the equal (unequal) option in condition c, for participant s and trial i; $\mathrm{E}^{O}_{(c,\,s,i)}$ $\left( \mathrm{I}^{O}_{(c,\,s,i)} \right)$ indicates the partners' payoff in the equal (unequal) option in condition c, for participant s and trial i.

The following free parameters were estimated for each participant and context separately:

$\alpha_{(c,\,s)}$: decision threshold, which indicates the amount of evidence required for making a decision (symmetric around zero);

$\beta_{(c,\,s)}$: starting point, which represents the initial bias towards equal or unequal option (0 = no bias);

$\kappa_{(c,\,s)}$: drift rate modulator, which scales the input of the subjective value difference between the equal and the unequal option (VD);

$\omega_{(c,\,s)}$: relative weight on others' payoff, which reflects the individual's concern for others' profits (i.e. altruistic preferences);

$\lambda_{(c,\,s)}$: leak strength, which adaptively controls acceleration or deceleration of evidence accumulation at the current time point (**Bogacz et al., 2006**; **Brunton et al., 2013**);

$\tau_{(c,\,s)}$: non-decision time (nDT), which accounts for sensory and motor processes not contributing to evidence accumulation per se, such as early visual processing and motor-response initiation.

Evidence accumulation starts with an initial value $EA_{(0)}$ equal to the starting point parameter ($\beta$). This value is then updated in discrete time steps of $dt = 0.001$ s until $|EA_{(t)}|$ is greater than the decision threshold parameter ($\alpha$). The model will make an equal decision when $EA_{(t)} > \alpha$, and make an unequal decision when $EA_{(t)} < -\alpha$. The noise of evidence accumulation at each time step is drawn from a Gaussian distribution of $N\left(0, \sigma\right)$, and we fixed $\sigma$ as 1.4. The subjective value differences between the equal option and the unequal option (VD) are constructed based on the Charness-Rabin utility model (**Charness and Rabin, 2002**), where $\mathrm{E}^{S}_{(c,\,s,i)}$ $\left( \mathrm{I}^{S}_{(c,\,s,i)} \right)$ indicates participants' payoff of the equal (unequal) option in condition c, participant s, and trial i, and $\mathrm{E}^{O}_{(c,\,s,i)}$ $\left( \mathrm{I}^{O}_{(c,\,s,i)} \right)$ indicates the partners' payoff of the equal (unequal) option in condition c, participant s, and trial i. Once the evidence ($EA_{(t)}$) reaches the decision threshold, the RT is calculated as $t \times dt + \tau$, where $\tau$ is the non-decision time.

We used the differential evolution algorithm introduced by **Mullen et al., 2011** to estimate the values of the free parameters described above separately for each participant and context. We ran the estimation with 60 population members over 150 iterations. For each iteration, we simulated 3000 decisions and RTs for VD in each trial for each participant with the free parameters for each population member. The likelihood of the observed data was computed given the distribution generated by the 3000 simulations for a certain set of parameters. Then the population evolves to a set of parameters maximizing the likelihood of the observed data with the procedures described by **Mullen et al., 2011**. We checked the evolution of the population across the 150 iterations and found that the algorithm reached a set of best-fitting parameters well before the 150 iterations in our data. The upper and lower bounds of each parameter are listed in **Supplementary file 4**. We selected uninformative bounds of these parameters based on previous studies (**Maier et al., 2020**; **Polanía et al., 2015**; **Polanía et al., 2014**). The bounds for the parameter of weight on others ($\omega$) were set from –1 to 1 because these are the limits inherent in the model (the sum of the weighting on others' payoff and one's own payoff is 1 in the Charness-Rabin model).

In the above OU model (full), we estimated all the parameters separately across contexts and participants. To confirm that this full OU model is good to explain individuals' behaviors, we fitted a series of alternative models for model comparison analyses. Specifically, first, we fit a drift-diffusion model (DDM) that kept the same parameters as the full OU model except for the leak strength parameter. Second, to confirm that the weight on others ($\omega$) and decision threshold ($\alpha$) were indeed different across contexts, we included models in which the weight on others ($\omega$) and/or decision threshold ($\alpha$) were constrained to a unique value across contexts as benchmark models. Model comparison results showed that the full OU model was a better fit than the other models to account for participants'

behavioral data (i.e. with the lowest BIC = 4058 ± 46 (Mean ± SE)). For details, see *Supplementary file 5*. The full OU model was the best fit for all 38 participants (100%). These results demonstrated that separating the parameters for leak strength, weight on others, and decision threshold across contexts did improve model fits while taking into account model complexity. In addition, we also estimated one OU model which used the first vs second choice (OU order model), rather than the equal vs unequal choice, as the decision boundaries. This model also correctly captured response speed and choice (*Figure 3—figure supplement 4*), and the estimated parameters (i.e. weight on others, drift rate, decision threshold, and non-decision time) were highly correlated between the OU order model and the winning model in both contexts (*Figure 3—figure supplement 5*). The BIC of the OU order model (4059±47) was slightly higher than the winning model. Since we are interested in the decision mechanism of individuals' altruistic choice, rather than the effect of offer presentation order, the OU order model was only considered as a validation of the winning model. Our study was not motivated by the question which specific decision model provides the best fit to our behavioral data, but rather by the question whether differences in altruism relate to well established model-predicted neural evidence accumulation processes that have documented in several prior studies (*Pisauro et al., 2017*; *Polanía et al., 2014*). For the purpose of our study, it is thus not important to examine the validity of other choice models that do not allow us to derive predictions for EEG signals. We thus do not claim based on our results that the OU model is the perfect model for altruistic decision making, but rather that it is a well-established tool to derive predictions for EEG signals related to evidence accumulation.

To evaluate model performance, we also calculated the sensitivity/specificity and balanced accuracy of the OU model. These measures are defined as follows:

$$Sensitivity_{(Equal)} = \frac{Number\ of\ correctly-predicted\ equal\ trials}{Number\ of\ observed\ equal\ trials}$$

$$Specificity_{(Unequal)} = \frac{Number\ of\ correctly-predicted\ unequal\ trials}{Number\ of\ observed\ unequal\ trials}$$

$$Balanced\ Accuracy = \frac{Sensitivity_{(Equal)} + Specificity_{(Unequal)}}{2}$$

Please note that while the model exhibited lower and more variable predictive accuracy for unequal (equal) choices in the DIS (ADV) contexts, this can be explained by how the different choices tap differentially into the individuals' variable preferences: In each context, one choice option is dominant, as evident by that finding that P(equal)=0.74 ± 0.006 in DIS and P(equal)=0.27 ± 0.009 in ADV. Participants thus more uniformly choose the dominant equal (unequal) choice option in the DIS (ADV) context and only people with stronger altruistic preferences choose the non-dominant unequal (equal) choice option, leading to larger variability. However, this asymmetry cannot have biased our results, since we only included a single weighting parameter to capture individuals' decision weight on other-payoff (i.e. altruistic preferences) in the model for each context, rather than having two different parameters to separately capture individuals' preferences for equality and inequality. It is also important to note that there is no difference in overall predictive accuracy between ADV and DIS conditions.

## EEG analysis

We performed EEG data analyses using Fieldtrip (*Oostenveld et al., 2011*) implemented in Matlab R2016b (MathWorks). We first identified eye movements and other noise artifacts using independent component analysis, and removed artifact components based on careful inspection of both topography and power spectrum of the components. We then used discrete Fourier transform to filter line noise. For response-locked analyses, epochs were extracted for a 1200ms time window around the response, ranging from 1000ms before to 200 after the response timepoint. Baseline corrections were performed by taking the duration of 200 ms to 0 ms before the second option onset as the baseline. For stimulus-locked analyses, we extracted epochs for a 1500ms time window around the second option onset, ranging from 500ms before to 1000ms after the stimuli onsets. Baseline corrections were performed by taking the duration of 200 ms to 0 ms before the second option onset as the baseline. Next, we visually inspected each individual trial and removed all trials with extremely high variance (e.g. muscle artifacts) from the data. We excluded three participants from the analysis due to excessive artifacts. For the remaining 38 participants, 21% ± 8.5% (Mean ± SD) of the trials were rejected.

## Response-locked EEG analysis

To validate evidence accumulation (EA) processes with EEG data, we used an analytical approach previously established in our lab (*Polanía et al., 2014*). We first simulated EA curves for each trial 500 times using each participants' best-fitting parameters and self-/other-payoffs in each trial and averaged these 500 simulated curves to compute a trial-specific model predicted EA trace.

For EEG data, we focused on a time window starting 600ms prior to the response and lasting until 100ms before the response. We excluded the last 100ms before the response to avoid motor execution-related signals due to abrupt increase in cortico-spinal excitability during this period. We examined the EEG-signal-associated EA processes using a cross-validation approach which combined both qualitative and quantitative criteria.

We divided trials into even and odd trials to test predictions of the EA model for each participant. We first used the even-numbered trials to identify channels where EEG signals were closely related to the shape of the model-predicted EA signal in each context, and then formally tested the model predictions against the data from the independent odd-numbered trials.

We carried out the analysis for the even-numbered trials in two steps: First, we calculated a linear regression with 250 time points (i.e. 600 ms to 100 ms in steps of 2ms before response in each trial) × N trials (i.e. number of trials in each context) between the ERP magnitude of even-numbered trials and model-predicted EA signals. Channels that survived Bonferroni correction at $p < 0.05$ were selected. Then, from the channels surviving the quantitative test, we further selected channels for which the average magnitude laid within 1 SE of the model predictions to ensure that the ERP signals were closely related to the EA model predictions across the full temporal interval.

Then, to test cross-validation predictions, we examined the relationship between the model-predicted ERP signals and actual ERP signals in the fully independent odd-numbered trials, based on the channels identified in the previous analysis.

To compare the relationship of model predictions and data between DIS and ADV contexts, we performed a *t* test of the difference between contexts in correlation ($r_{Pearsons}$) between model predictions and data for each EEG channel.

To account for multiple comparisons, we identified significant clusters based on two-sided *t* statistics with a threshold of $p < 0.05$ at channel level and at least three significant neighbouring channels in a single cluster. For each cluster that survived the threshold, the size of the cluster was defined as the integral of the t scores across all the channels of the cluster and its significance was tested with a permutation statistic (i.e. performing iterations 5000 times with shuffled context labels to generate a distribution of cluster sizes for the null hypothesis of no difference between contexts) (*Maris, 2012*). Significant clusters surviving the cluster correction for multiple comparisons at $p < 0.05$ are reported. We applied this threshold for all the whole-brain EEG analyses to identify significant clusters in the current study.

To calculate the built-up rate, we followed Kelly and O'Connell's work (*Kelly and O'Connell, 2013*) and extracted grand average response-locked ERP waveforms from the identified clusters in previous analysis. Then, we fitted a linear slope to the waveforms in the time window from –300ms to –100ms before response and took the steepness of these slopes as indices of build-up rates for each participant.

## Stimulus-locked EEG analyses

With stimulus-locked EEG analyses, we aimed to identify channels and time windows in which neural activity was associated with self- and other-interest (i.e. self-/other-payoff change between the second and the first option) processing in each context. For each channel, each epoch was integrated over 40ms windows from 0 to 1000ms after the second option onset, generating a matrix of EEG signals in 128 × 25 time windows for each trial. Then, we used the EEG signals across all the trials in each context in a linear regression model described as follows:

$$y_{(channel, time)} = \beta_{0(c, s)} + \beta_{1(c, s)} \times \Delta S_{(c, s)} + \beta_{2(c, s)} \times \Delta O_{(c, s)} + \varepsilon$$

where $y_{(channel, time)}$ is a matrix of trial-by-trial data for each channel and each time window, $\Delta S$ is the matrix of standardized participants' own payoff changes from the first to the second option (second minus first) across all the trials, $\Delta O$ is the matrix of standardized partners' payoff changes from the first to the second option across all the trials, $\varepsilon$ is the error term. This linear regression produced

a set of estimated coefficients (i.e. $\beta_1$ and $\beta_2$ maps) measuring the parametric effect strengths of $\Delta S$ and $\Delta O$ on neural activity for each channel and time window in each context and participant ($c$ for condition, and $s$ for participant).

To investigate how the neural weighting of $\Delta S$ and $\Delta O$ varied across DIS and ADV contexts, we contrasted $\beta_1$ ($\beta_2$ maps between the two contexts by performing paired-sample $t$-tests over all participants. We report the results based on permutation tests to account for multiple comparisons as described above. Significant clusters surviving the cluster correction for multiple comparisons at p < 0.05 are reported.

To further investigate whether and how the neural weighting of $\Delta S$ and $\Delta O$ varied across individuals with different altruistic preferences, we split participants into more altruistic (MA) and less altruistic (LA) groups by the median of the weight on others parameter or of the starting point, derived from the OU model in each context, respectively. For each context, we contrasted $\beta_1$ ($\beta_2$) maps between the two groups by performing independent-sample t-tests. To visualize the effects of significant clusters and to investigate EEG signals associated with the processing of different dimensions of information (i.e., $\Delta S$ and $\Delta O$), we further split trials in each context into trials with positive and negative $\Delta S$ or trials with positive and negative $\Delta O$ and extracted average ERP waveforms for each of these sub-conditions. For the interaction analyses of payoff change and context over ERP waveforms, we also report the results based on permutation tests to account for multiple comparisons as described above. Only significant clusters surviving the cluster correction for multiple comparisons at p < 0.05 are reported.

## Connectivity analysis

To investigate the neural communication characteristics between channels associated with the processing of other-payoff and channels associated with evidence accumulation processes, we performed connectivity analyses using the debiased weighted phase lag index (dWPLI) (*Vinck et al., 2011*). This coherence statistic is based only on the imaginary component of the cross-spectrum, which ensures that potential differences in power between the contexts or groups do not affect direct cross-context or cross-group comparisons of coherence. It also removes the type of bias due to the small sample size in the estimation of the phase lag index and is less sensitive to noise and volume conduction.

We first performed spectral estimates (both power and cross-spectral density) for each response-locked epoch using a multi-taper method implemented in Fieldtrip. The time-frequency analyses were performed in the frequency range between 16 and 100 Hz. The length of the temporal sliding window was eight cycles per time window in steps of 0.02 s. The width of frequency smoothing was set to 0.3× f with a frequency resolution in steps of 1 Hz. We then calculated the dWPLI of all possible pair-wise connections between the $\Delta O$ processing channels in DIS shown in *Figure 5A* and the EA-related channels in DIS shown in *Figure 2C* for each participant. We focused on the signals ranging between 600ms and 100ms before responses at gamma band frequency (30 Hz to 90 Hz).

To compare the dWPLI between MA and LA groups, we conducted an independent-sample t-statistic of dWPLI differences between groups for each time step. We thresholded the t-statistic map with p < 0.05. For each cluster surviving this threshold, the size of each cluster was defined as the integral of the t scores (group effect) across all channels of the cluster and its significance was tested with a permutation statistic, that is, the cluster identification was repeated 5000 times by shuffling group labels to generate a distribution of cluster sizes for the null hypothesis of no difference between groups. Only significant clusters surviving the cluster correction for multiple comparisons at p <0.05 are reported.

## Acknowledgements

This project has received funding from the European Research Council (ERC) under the European Union's Horizon 2020 research and innovation programme (grant agreement No 725355, ERC consolidator grant BRAINCODES). Christian C Ruff received funding from the University Research Priority Program 'Adaptive Brain Circuits in Development and Learning' (URPP AdaBD) at the University of Zurich

# Additional information

## Funding

| Funder | Grant reference number | Author |
|---|---|---|
| European Research Council | European Union's Horizon 2020 research and innovation programme, No: 725355 | Christian C Ruff Jie Hu Arkady Konovalov |
| University of Zurich | University Research Priority Program 'Adaptive Brain Circuits in Development and Learning' (URPP AdaBD) at the University of Zurich | Christian C Ruff |

The funders had no role in study design, data collection and interpretation, or the decision to submit the work for publication.

## Author contributions

Jie Hu, Conceptualization, Data curation, Formal analysis, Investigation, Visualization, Methodology, Writing – original draft, Project administration, Writing – review and editing; Arkady Konovalov, Conceptualization, Data curation, Investigation, Methodology, Writing – original draft, Writing – review and editing; Christian C Ruff, Conceptualization, Resources, Supervision, Funding acquisition, Investigation, Visualization, Methodology, Writing – original draft, Project administration, Writing – review and editing

## Author ORCIDs

Jie Hu  http://orcid.org/0000-0002-0991-9254
Arkady Konovalov  http://orcid.org/0000-0002-9448-6659
Christian C Ruff  http://orcid.org/0000-0002-3964-2364

## Ethics

Participants were informed about all aspects of the experiment and gave written informed consent. The experiment conformed to the latest revision of the Declaration of Helsinki, and the protocol was approved by the Ethics Committee of the Canton of Zurich (KEK: 2011-0239/3).

## Decision letter and Author response

Decision letter https://doi.org/10.7554/eLife.80667.sa1
Author response https://doi.org/10.7554/eLife.80667.sa2

# Additional files

## Supplementary files

- MDAR checklist

- Supplementary file 1. Generalized linear mixed-effects model results of choice data.

- Supplementary file 2. Generalized linear mixed-effects model results of choice data show that the presentation order (i.e., first or second) of the more equal or unequal option does not bias individuals' choices.

- Supplementary file 3. Linear mixed-effects model results of RT data.

- Supplementary file 4. Bounds of OU parameters.

- Supplementary file 5. Model comparison results.

## Data availability

All relevant data (behavioral and EEG) and customized Matlab and R codes are available online (https://doi.org/10.17605/OSF.IO/KPGSA).

The following dataset was generated:

| Author(s) | Year | Dataset title | Dataset URL | Database and Identifier |
|---|---|---|---|---|
| Hu J, Konovalov A, Ruff C | 2022 | A unified neural account of contextual and individual differences in altruism | https://doi.org/10.17605/OSF.IO/KPGSA | Open Science Framework, 10.17605/OSF.IO/KPGSA |

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

## Appendix 1

### Behavior: Choice depends differentially on self- and other-payoffs across contexts

To visualize relationships between the effect of self-payoff, other-payoff, and inequality context on choice outcome identified in model-free linear mixed-effects regressions, we divided trials for each participant and context into seven bins based on the z-score of the Self-payoff Change ($\Delta S$). Then we further split trials in each bin into trials in which the second option would benefit or harm the other financially, relative to the first option (i.e. Increased other-payoff (OP) when the second option profits the other versus Decreased other-payoff (OP) when the second option leads to lower payoff for the other relative to the first option, *Figure 1—figure supplement 2A* left and middle panels). We then fitted psychometric functions to these data and compared their slopes to determine how sensitive choices were to changes in self-payoff change in the two inequality contexts, and how this differed between choices that increased versus decreased the other person's payoff from the first to second option (*Figure 1—figure supplement 2A* left and middle panels). For choices that would decrease the other's payoff, participants reacted less strongly to the associated possible increases in self-payoff in the advantageous context (lower slope of Self-payoff Change function in ADV than DIS for Decreased Other-payoff (OP) trials; 1.74±0.12 vs 2.02±0.10 (MEAN ± SE), ADV vs DIS: 95% CI [- 0.59–0.03], Cohen's d = - 0.41, *t*(37) = –1.85, p = 0.07). By contrast, for choices that would increase the other's payoff, people were equally sensitive to increases in self-payoff between both inequality contexts (Increased Other-payoff (OP); ADV: 2.09±0.12, DIS: 2.03±0.11, ADV vs DIS: 95% CI [- 0.20–0.31], Cohen's d=0.07, *t*(37) = 0.45, p = 0.65; direct comparison of both effect: *F*(1,37) = 3.54, p = 0.068, *Figure 1—figure supplement 2A* right panel). Model simulations based on the winning SSM model also showed that for choices that would decrease the other's payoff, participants reacted less strongly to the associated possible increases in self-payoff in the advantageous context (lower slope of Self-payoff Change function in ADV than DIS for Decreased Other-payoff (OP) trials; 1.37±0.06 vs 1.64±0.09, ADV vs DIS: 95% CI [- 0.44 – - 0.09], Cohen's d=-0.51, *t*(37) = –3.14, p = 0.003). For choices that would increase the other's payoff, people also reacted less strongly to the associated possible increases in self-payoff in the advantageous context, but with a smaller effect (Increased Other-payoff [OP]; ADV: 1.45±0.07, DIS: 1.73±0.10, ADV vs DIS: 95% CI [- 0.49–0.07], Cohen's d=-0.44, *t*(37) = -2.69, p = 0.011). Taken together, these results confirm that our SSM model predicts the behavioral pattern observed in the data, providing evidence for the correspondence between the model-free and the model-based analysis.

### Behavior: Presentation order of options does not affect equal/unequal choices

To test whether properties of the reference option would affect individuals' choice, we ran a linear mixed-effects model in which individuals' equal/unequal choice (equal = 1, unequal = 0) was the dependent variable and the other-payoff difference between the equal/unequal option and the advantageous/disadvantageous context were independent predictors. Since the self-payoff difference between the equal and unequal option was negatively correlated with other-payoff difference, we did not include it in this control analysis. Importantly, we also included a predictor indicating whether the equal option was the reference (0) or the alternative option (1). This reference option indicator was not a significant predictor for individuals' equal choice (*Supplementary file 2*). Therefore, the asymmetry of the reference/alternative option or the presentation order of options did not affect participants' equal/unequal choice.

### ERP results

To investigate how the brain processes payoff information, we analyzed ERPs related to different aspects of the payoff information (e.g., self-payoff/other-payoff changes between the second option and the first option) upon stimulus presentation. In the analyses reported in the main text, we have identified two different spatial-temporal clusters showing opposite effects of self-payoff changes between different inequality contexts (i.e., earlier effect: from ~240–360ms after stimulus onset, *Figure 4A*; later effect: from ~440–800ms after stimulus onset, *Figure 4C*).

To further clarify whether the time courses for neural processing of self-payoffs were different or similar in the two contexts, we first directly compared the ERP components corresponding to the effects identified in the above analyses. For this analysis, we categorized trials in each context into trials in which the second option increases self-payoff (i.e., trials with more equal second option in

DIS and more unequal second option in ADV) and trials in which the second option decreases self-payoff (i.e., trials with more unequal second option in DIS and more equal second option in ADV), and examined average ERP waveforms for each level of self-payoff change and context. Second, we performed interaction analyses between context (ADV vs. DIS) and self-payoff change (Increased SP vs. Decreased SP) over average ERP signals in the time window of ~440–800ms after stimulus onset identified in the initial regression analysis.

In the following two sections, we reported detailed statistical information for these stimuli-locked ERP results.

## Different ERP responses are associated with self-interest in different contexts

By directly comparing ERP dynamics corresponding to the effects identified in the regression analyses reported in the main text, we confirmed that for both time windows (i.e., the early window:~240–360ms after stimulus onset, and the late window:~440–800ms after stimulus onset), ERPs associated with the increase and decrease of self-payoff yielded opposite neural effects in the DIS and ADV contexts (see *Figure 4B and D*). Specifically, factorial analysis of ERPs revealed that for the earlier time window (~240–360ms after stimulus onset), in the ADV context, self-payoff increases (–0.75±0.15 μV, Mean magnitude ± SE, 95% CI [-1.04,–0.46], Cohen's d=–0.84, $t(37) = -5.17$, p < 0.001) evoked a stronger negative response than self-payoff decreases (–0.59±0.13 μV, 95% CI [-0.86,–0.33], Cohen's d=–0.74, $t(37) = -4.59$, p < 0.001) at trend level (Increased SP vs. Decreased SP: 95% CI [–0.32, 0.01], Cohen's d=–0.31, $t(37) = -1.91$, p = 0.06). Similarly, in the DIS context, self-payoff decreases (–0.66±0.15 μV, 95% CI [-0.95,–0.36], Cohen's d=–0.72, $t(37) = -4.46$, p < 0.001) evoked a more negative response than self-payoff increases (–0.54±0.13 μV, 95% CI [-0.80,–0.29], Cohen's d=–0.70, $t(37) = -4.29$, p < 0.001) also at trend level (95% CI [–0.24, 0.02], Cohen's d=–0.28, $t(37) = -1.70$, p = 0.098, *Figure 4B*). In the later time window (~440–800ms after stimulus onset), stronger positive responses were seen for self-payoff increases (1.05±0.11 μV, 95% CI [0.83, 1.26], Cohen's d=1.59, $t(37) = 9.81$, p < 0.001) than self-payoff decreases (0.86±0.07 μV, 95% CI [0.72, 1.01], Cohen's d=1.97, $t(37) = 12.17$, p < 0.001) in ADV (Increased SP vs. Decreased SP: 95% CI [0.04, 0.32], Cohen's d=0.43, $t(37) = 2.64$, p = 0.012), whereas stronger positive responses were observed for self-payoff decreases (0.95±0.09 μV, 95% CI [0.77, 1.13], Cohen's d=1.75, $t(37) = 10.80$, p < 0.001) than self-payoff increases (0.76±0.08 μV, 95% CI [0.61, 0.92], Cohen's d=1.61, $t(37) = 9.92$, p < 0.001) for DIS (Decreased SP vs. Increased SP: 95% CI [0.07, 0.30], Cohen's d=0.53, $t(37) = 3.26$, p = 0.002, *Figure 4D*).

## Temporal difference of self-interest processing across inequality contexts

For the time window of ~440–800ms after stimulus onset, we observed an earlier neural response to the self-payoff change in ADV (~450–600ms after stimulus onset) than in DIS (~630–700ms after stimulus onset, *Figure 4D and E*). To visualize the interactive effects of self-payoff change and context in these two windows of interest (WOIs), we extracted the ERP data from each WOI. The comparison showed that for the early WOI, self-payoff increases evoked stronger neural responses (1.04±0.10 μV, 95% CI [0.84, 1.24], Cohen's d=1.66, $t(37) = 10.26$, p<0.001) than self-payoff decreases (0.71±0.07 μV, 95% CI [0.57, 0.85], Cohen's d=1.68, $t(37) = 10.38$, p<0.001) in ADV (Increased SP vs. Decrease SP: 95% CI [0.18, 0.49], $t(37) = 4.44$, Cohen's d=0.72, p<0.001). And, self-payoff decreases evoked stronger neural responses (0.76±0.09 μV, 95% CI [0.58, 0.94], Cohen's d=1.42, $t(37) = 8.73$, p<0.001) than self-payoff increases (0.64±0.07 μV, 95% CI [0.50, 0.78], Cohen's d=1.44, $t(37) = 8.91$, p<0.001) in DIS (Decreased SP vs. Increase SP: 95% CI [0.02, 0.23], $t(37) = 2.44$, Cohen's d=0.40, p = 0.04, *Figure 4E* left panel). For the late WOI, self-payoff decreases (1.12±0.11 μV, 95% CI [0.90, 1.34], Cohen's d=1.63, $t(37) = 10.07$, p<0.001) evoked stronger neural responses than self-payoff increases (0.79±0.09 μV, 95% CI [0.61, 0.97], Cohen's d=1.42, $t(37) = 8.76$, p<0.001) in DIS (Decreased SP vs. Increased SP: 95% CI [0.17, 0.47], Cohen's d=0.69, $t(37) = 4.26$, p<0.001), but there were no such differences in neural responses between self-payoff decreases (0.87±0.10 μV, 95% CI [0.66, 1.07], Cohen's d=1.36, $t(37) = 8.36$, p<0.001) and increases (0.99±0.12 μV, 95% CI [0.75, 1.23], Cohen's d=1.31, $t(37) = 8.10$, p<0.001) in ADV (Decreased SP vs. Increased SP: 95% CI [–0.31, 0.06], Cohen's d = - 0.22, $t(37) = -1.36$, p = 0.36, *Figure 4E* right panel).

## Visualization for the relationship between individual differences in altruistic preferences and neural processing of other-payoffs

To visualize the specific origin of the neural effect identified in the individual-difference analysis of other-payoff processing, we tested the sign and time course of the corresponding ERP components in the two groups. This revealed that less altruistic participants responded with a more negative early ERP component to increases in other-payoff. To perform this analysis, we categorized trials in the DIS context into trials in which the second option increased other-payoff (Increased OP) and trials in which the second option decreased other-payoff (Decreased OP) and extracted average ERP waveforms for each other-payoff change level and group. We found that in the less altruistic group, ERP waveforms were negatively correlated with the increase of other-payoff (Increased OP: $-0.39\pm0.20$ μV, 95% CI [$-0.81$, 0.02], Cohen's d$=-0.46$, $t(18) = -2.00$, p = 0.06), and this correlation was significantly stronger than the corresponding correlation (Decreased OP: $-0.07\pm0.18$ μV, 95% CI [$-0.44$, 0.31], Cohen's d$=-0.08$, $t(18) = -0.36$, p = 0.72) with the decrease of other-payoff (Increased OP vs Decreased OP: 95% CI [-0.54,$-0.12$], Cohen's d$=-0.77$, $t(18) = -3.36$, p = 0.004). For the more altruistic group, no such effects were observed (Increased OP: $-0.06\pm0.17$ μV, 95% CI [$-0.41$, 0.30], Cohen's d$=-0.08$, $t(18) = -0.33$, p = 0.75; Decreased OP: $-0.14\pm0.16$ μV, 95% CI [$-0.48$, 0.20], Cohen's d$=-0.20$, $t(18) = -0.89$, p = 0.39; Increased OP vs Decreased OP: 95% CI [$-0.06$, 0.23], Cohen's d$=0.29$, $t(18) = 1.25$, p = 0.23, *Figure 5—figure supplement 4*).

## Exploratory analyses of alternative scenarios

### Neural processing of the first option and the interval leading up to the second option

Since we sequentially presented the two alternative options in each trial, one may wonder how people evaluate the payoffs in the first option, and whether evaluation of the first option could affect the starting point bias in the evidence accumulation process after the presentation of the second option. To address these issues, we performed similar GLM analyses as for the main analyses. In one GLM, we included the value of the self- and other-payoffs in the first option as regressors, whereas in another GLM model, we included the difference between self- and other-payoffs in the first option as the regressor to predict ERP signals. In the first GLM, we did not observe any significant ERP cluster in which neural responses were related to the magnitude of self-/other-payoffs or self- vs other-payoff difference in either inequality context. In the second GLM, there were no significant ERP responses differently associated with self-/other-payoff or self- vs other-payoff difference across the two contexts. For both models, there was no significant ERP cluster in which the responses to self-/other-payoffs or self- vs other-payoff differences were related to the starting point bias derived from the winning model or the OU order model in either context.

Nevertheless, it may still be plausible that neural responses right before the presentation of the second option are associated with individuals' starting point bias. To test this possibility, we re-analyzed the ERP responses locked to the presentation of the second option by taking 500–200ms before the second option onset as the baseline period to re-perform the baseline correction. Then, we conducted the same GLM analyses as above for the interval from –200–0ms before the second option onset. We also correlated the starting point bias in either the winning OU model or the OU order model with the neural responses related to the self-payoff, other-payoff, or the self- vs other-payoff difference in the first option or the raw ERP signals in either context. These analyses also did not reveal any significant cluster in which the pre-stimulus ERP responses were related to the starting point bias across participants.

We obviously cannot draw any firm conclusions based on the above null results. However, our results do suggest that the value-comparison and evidence-accumulation processes after the presentation of the second option are more informative with respect to neural origins of individual and contextual differences in altruism than any evaluation process at the time of the presentation of the first option.

### Neural correlates related to integrated value

To explore whether and how ERP signals reflect how people represent integrated value (e.g. utility difference between chosen and unchosen option, as often focussed on in fMRI studies, see *Bartra et al., 2013*), we performed GLM analyses by taking the integrated value as the regressor to predict ERP signals after the second option onset. The analysis did not reveal any significant responses related to the integrated value within each context or any responses showing differences between

the two contexts. These null results are not surprising since the integrated values are usually encoded in the ventromedial prefrontal cortex (*Bartra et al., 2013*), a deep brain region for which signals may not be easily detected by EEG.

## Neural correlates related to individual differences in altruism

In addition, we also explored the neural correlates of individual differences by correlating weight on others ($\omega$) with ERP signals related to the self-payoff change in each context, and correlating differences in $\omega$ between contexts with the difference in ERP responses related to self-payoff change between contexts. These analyses also did not reveal any significant cluster.

Since both the ERP responses related to other-payoff change during 320–400ms after second option onset and the frontal-parietal dWPLI during 520–460ms before response are correlated with individuals' weight on other others ($\omega$), we are also interested in identifying whether the frontal-parietal dWPLI functions as a mediator to account for the effect of ERP responses to the other-payoff change on individuals' altruistic preferences ($\omega$). Therefore, we first performed correlation analyses, and found that the ERP responses to other-payoff change (320–400ms after second option onset) were not significantly correlated with frontal-parietal dWPLI strength (520–460ms before response) across participants (Pearson $r=0.18$, p = 0.28; Kendall tau = 0.001, p = 1.00). The mediation model (with the ERP responses as the independent variable, dWPLI strength as the mediator variable, and the parameter of weight on other others' payoff ($\omega$) as the dependent variable) could neither be established as the path from ERP responses to dWPLI strength was not significant, and the mediation model as a whole was not significant as well. Therefore, we do not further speculate about the relationship between ERP responses and dWPLI strength. Note that, although the ERP responses during 320–400ms after the second option onset and frontal-parietal dWPLI during 520–460ms before response temporally overlapped with each other (see main text), these two types of signals actually reflect different neural activity characteristics (i.e., ERP responses reflect postsynaptic potential, while dWPLI reflect interregional oscillatory phase coupling), which may make it harder to establish a direct relationship between them.

