## [Editor Report]

In this important paper, the authors use a sophisticated combination of computational modeling and EEG to show that variation in generosity produced by changes in context (i.e., disadvantageous vs. advantageous inequality) and variation due to individual differences in concern for others both seem to occur early, during the perceptual or valuation stage of a choice, rather than later on during choice comparison. However, these two sources of variation also appear to operate through distinct mechanisms during this stage of processing, which spurs further questions about the drivers of human prosocial behavior. This paper will be of considerable interest to researchers studying the psychological and neural basis of variation in prosocial behavior.

---

## [Decision Letter]

**Decision letter after peer review:**

Thank you for submitting your article "A unified neural account of contextual and individual differences in altruism" for consideration by *eLife*. Your article has been reviewed by 2 peer reviewers, including Redmond G O'Connell as the Reviewing Editor and Reviewer #1, and the evaluation has been overseen by Michael Frank as the Senior Editor. The following individual involved in review of your submission has agreed to reveal their identity: Cendri A Hutcherson (Reviewer #2).

Essential revisions:

1) Both reviewers indicate that the framing of the paper should be improved to provide greater clarity regarding the key study questions and hypotheses with greater reference to the prior literature. Currently the study is framed such that a lot of emphasis is placed on the question of whether or not there are common mechanisms for altruistic decisions in different contexts yet it seems a forgone conclusion that some common mechanisms will exist when both types of decisions studied in this task involve the same motor response.

2) Dig more deeply into the concordance or discordance between EEG and modeling results. For example, there are apparent inconsistencies between the modelling results which indicate context-dependent bound adjustments and the ERP data which do not seem to show any such effect that are not discussed by the authors. Elsewhere, while participants seem to put more weight on others' payoff changes in the advantageous inequality context, there are no EEG signals that encode changes in other outcomes as a main effect in the advantageous context, and it is individual variation in encoding of others' payoffs in the disadvantageous context that relate to individual differences in equality-seeking in that context.

*Reviewer #1 (Recommendations for the authors):*

The modelling indicates context-dependent differences in decision bounds but the analyses of the response-locked EEG evidence accumulation signals indicates no such differences. I would have thought that a high model-estimated bound should correspond to a larger signal amplitude at response. Can the authors account for this apparent discrepancy?

Can the authors provide an intuition for why a leaky accumulator model was the best fitting in the case of a task in which the evidence is fixed throughout the trial.

Lines 276-280. The authors indicate that the fact that choice RTs are shortened on easier trials is indicative of an evidence accumulation process informing decisions. In fact, several prominent models that invoke little (e.g. Urgency Gating) or no (e.g. Extrema) would also predict faster choices when evidence is stronger.

321-322 – the 'more technical parameters' leak and NDT. Please explain what is meant by 'more technical'.

401-402 – NDT as speed of pre-decision processes – this should be changed to read speed of pre- and post-decision processes as NDT also include motor execution delays.

*Reviewer #2 (Recommendations for the authors):*

This is an interesting and carefully designed study on altruistic choice that applies rigorous computational modeling methods and observes intriguing correlates of contextual and individual difference influences on behavior. I think it deserves to be published in *eLife*, and in general think it is a well done paper.

In your introduction, you frame the questions your paper will address around determining whether altruistic choice is accomplished via fundamentally different mechanisms in different contexts and across different individuals. I found myself wondering whether you could make this clearer by emphasizing the distinction between qualitative differences (i.e., process level differences) and merely quantitative differences (i.e., differences in sensitivity of a process to a quantity, but not in the fundamental stages of processing) and whether it might help to have a figure that illustrates the different possibilities and different stages at which one might expect differences.

Lines 125-126 of the introduction: you point to a question of whether differences reflect involvement of hard-wired choice mechanisms. I'm not sure any of your data can speak to whether this is hard-wired, and this doesn't really seem to be the focus of your analysis anyway, so I would probably suggest avoiding this as a focus in your intro.

Lines 304-306: could you actually spell out what the winning model was here in the results, to save the lazy reader from having to dig into methods and SI Materials to figure this out?

Not critical, but it is interesting to me that in the GLM analysis, you find a significant interaction effect between self and other payoff. That doesn't show up in your O-U model, if I understand correctly, or in any of your subsequent analyses. Is this interaction predicted by Charness-Rabin, or by your EEG results?

What are we to make of the fact that individual differences in parameters are strongly correlated across the contexts, but the EEG signals seem to be categorically different? This might be worth explicitly addressing in the Discussion section.

Was there any correlate of the difference in decision bound between ADV and DIS contexts across participants? In Harris et al. (2018), a correlate of decision bound was observed in response-locked data. I found myself wondering whether you might observe something similar here.

Similarly, was there any correlate of the starting bias parameter? I am generally not a huge fan of DDM models in which the decision bounds do not actually map on to the responses subjects make, as you do here with your equal/unequal choice bounds, because they imply that subjects somehow precognitively know whether the first or second option will be the more equal and set their starting bias accordingly. Conceptually this makes no sense, and I am on record (Teoh et al., 2020) as arguing that this starting bias could result from post-stimulus attention and value accumulation mechanisms rather than pre-stimulus biases. Acknowledging that you find a very tiny BIC advantage for your model over first-second option bound model (though it's not clear to me whether that small difference could arise due to chance), and that the results end up being pretty similar across the two models for the parameter you focus on (i.e., omega), I found myself wondering whether the EEG signals might be useful here. For example, if you looked for an EEG correlate of starting bias in the period just before presentation of the second option, and correlated that with starting bias (either in the equal/unequal or first/second bound models), this might be another way of bolstering your message that most of the computations that influence choices/individual differences happen not at the late accumulation stage, but at the early perception and valuation stages.

Can you say more about what it means that the ERP-behavior correlation with δ-other in the DIS context is observed from 320-440ms stimulus-locked, but the phase coupling correlation is observed in response-locked data from 520-460ms prior to choice? Do these periods actually overlap? How do you see these signals as related if they occur in different time periods?

Is there any signal that correlates with overall integrated value/relative value of the second option? Fronto-central signals are often seen in studies of food choice from between 400-500ms after stimulus onset, for instance. Is something similar observed here during presentation of the second option? If so, how does that signal compare spatially and temporally to the signals related to δ-self and δ-other that you see here?

One of the most interesting or unexpected findings, to me at least, is the lack of any consistent signals for self and other across the two contexts, at least during presentation of the second option. While you acknowledge in your Discussion section that some of this may have to do with your experimental design, I found myself wondering whether you could leverage your design a bit more to provide greater insight into how this might play out, computationally. For example, the wheel-and-spoke design of your payoffs (Figure 1-supplement 1) means that, once a person knows the values of the first option for self and other, they also known within a narrow range what the values of the second option can possibly be. While it is only this relative value that matters for choice, it is clear that something occurring in response to the first option seems to 'set' the way that people process the second option. This could be what drives differences in starting point bias, as well as differences in response to δ-self/δ-other. You don't report any analyses performed during the time period of the first option, but I found myself wondering whether this might not be an extremely interesting time period to explore. For instance, do you see any signal that correlates with the overall magnitude of self or other during this period, time-locked to stimulus onset? Do you see a signal that differentiates ADV from DIS inequality contexts? Does that signal then predict on an individual basis how people respond to presentation of the second option? You already have a lot of analyses packed into this paper, so I'm not sure exactly that I am suggesting you include a whole new section on the first stage of your paradigm in this paper, but to be honest I found myself wondering whether that dynamic interaction between first and second-stage calculations might not end up being a more interesting and revealing place to explore, and yield a more interesting message about what predisposes people to act generously or selfishly, or to attend to self or other outcomes.

If I understood correctly, when you look at individual difference correlates, you only look at correlates of omega within each context separately. But it seems to me that another way of looking at this is via two separate individual differences: difference in overall omega (averaged over the two contexts) and differences in change in omega across the two contexts. I would be particularly interested in whether any of the changes in the signals you observe across context related to δ-self are related to omega-difference.

Is it possible to do a similar dWLPI analysis for the sensors that were differentially sensitive to δ-self in the two contexts? One might speculate that if connectivity between perceptual/value signals and evidence accumulation processes is a key mechanism for information transfer, then one should see an analogous result when looking at these sensors. Especially given that you don't observe any amplitude differences that correlate with omega, it might be that such differences are more likely to arise from information transfer differences.

On a similar vein: you find that both ERP sensitivity to δ-other and dWLPI predict differences in generosity/differences in omega in the DIS context. Is one of these more important, or mediating the other? For example, if information contained in ERPs is being transmitted via neural coupling, one might expect that the dWLPI either mediates or moderates the influence on choice of individual differences in ERP sensitivity.

In lines 813-818, you suggest that early signals (240-360ms post-stimulus) related to inequality processing, and that later signals (440-800) relate to the conflict between inequality level and self-/other-payoff change. I couldn't quite tell whether you are trying to argue that these signals are qualitatively different or not, and if so, on what basis you conclude that. The signals look fairly similar to me in terms of what they are sensitive to, so it wasn't obvious to me that one can conclude they are doing different things. Can you clarify?

In lines 834-837 of the conclusion, you suggest that the different temporal windows of sensitivity to self payoff change might account for the differences in reaction times. But your modeling results suggest that much of this difference could be due to changes in threshold. This probably deserves acknowledgement.

---

## [Author Response]

Essential revisions:1) Both reviewers indicate that the framing of the paper should be improved to provide greater clarity regarding the key study questions and hypotheses with greater reference to the prior literature. Currently the study is framed such that a lot of emphasis is placed on the question of whether or not there are common mechanisms for altruistic decisions in different contexts yet it seems a forgone conclusion that some common mechanisms will exist when both types of decisions studied in this task involve the same motor response.

We thank both reviewers and the editor for pointing us to this conceptual issue with how the first version of this manuscript study was framed. In the revised introduction, we have followed the reviewers’ suggestions and now emphasize the more specific question of our study: How are contextual and individual differences in altruism linked to neurocognitive mechanisms that are engaged during temporally different stages of the decision process (Lines 99-107 and 117-138)? We have also rephrased all other related sections throughout the paper along similar lines.

In addition, we have more explicitly tied our hypotheses to findings from previous studies as follows: We explain in more detail that different types of decisions that are reported via the same manual actions can still involve fundamentally distinct evidence accumulation processes. This has been shown in EEG studies that have directly compared matched perceptual and value-based decision making (and find only parietal accumulation signals for perceptual decision making vs. frontal and parietal accumulation signals for value-based decision making (Polanía et al., 2014)). Moreover, we highlight that altruistic decisions that have identical motor responses but are guided by different motives or preferences have also been found to involve activity in different brain networks (Hein et al., 2016). Based on these findings, we motivate the main question of our manuscript: Do individual and situational differences in altruism reflect differences in early information processing, value computations, or evidence accumulation processes, which occur at different timepoints of the decision process (Harris et al., 2018)? (Lines 180-201). We hope you concur that these changes have clarified the conceptual approach taken in our study.

2) Dig more deeply into the concordance or discordance between EEG and modeling results. For example, there are apparent inconsistencies between the modelling results which indicate context-dependent bound adjustments and the ERP data which do not seem to show any such effect that are not discussed by the authors. Elsewhere, while participants seem to put more weight on others' payoff changes in the advantageous inequality context, there are no EEG signals that encode changes in other outcomes as a main effect in the advantageous context, and it is individual variation in encoding of others' payoffs in the disadvantageous context that relate to individual differences in equality-seeking in that context.

We agree that our results are multi-faceted and that it is interesting to further examine the correspondence between the modelling and EEG results. Following both reviewers’ suggestions, we have carried out several additional analyses and have reported their results in the revised manuscript.

First, we investigated the correspondence between decision bounds and response-locked EEG signals. We found that the ERP amplitude at response (in the time window of 100 to 0 ms before response) is indeed higher in the DIS context compared to the ADV context. This is fully consistent with the sequential sampling model results that show higher decision thresholds in the DIS compared to ADV context. We have included this finding in the Results section (Lines 396-402 and Figure 2 —figure supplement 1).

Second, we performed more analyses to explore other potential scenarios suggested by both reviewers. We report these either in the main text (Lines 595-601, 678-682, 759-762) or in Appendix 1. For those analyses that have confirmed putative discordances between modelling and EEG results, we have included a more detailed discussion of the possible reasons for these discrepancies. We also acknowledge that more in-depth studies (e.g., simultaneous EEG-fMRI studies) may be needed to provide a complete picture of the origin of contextual and individual differences in altruism (Lines 934-941 and 982-1018).

We hope you agree that the revised manuscript now provides a comprehensive overview and discussion of the relation between the model-based and model-free results.

Reviewer #1 (Recommendations for the authors):The modelling indicates context-dependent differences in decision bounds but the analyses of the response-locked EEG evidence accumulation signals indicates no such differences. I would have thought that a high model-estimated bound should correspond to a larger signal amplitude at response. Can the authors account for this apparent discrepancy?

We thank both reviewers for pointing this out. We now investigate the correspondence between decision bounds and response-locked EEG signals. We re-analyzed the response-locked ERP signals and found that ERP amplitude in the time window of -100 to 0 ms before response is higher in the DIS than ADV context, which is entirely consistent with our sequential sampling modeling results showing higher decision thresholds in the DIS compared to the ADV context (Lines 396-400).

Can the authors provide an intuition for why a leaky accumulator model was the best fitting in the case of a task in which the evidence is fixed throughout the trial.

Although the payoffs of each option and thus the decision evidence were fixed throughout the trial, participants had to provide their response within a limited time (i.e., 3 s). This may have led to a strategic acceleration in evidence accumulation speed for later stages when the decision process approached the boundary. This strategic acceleration may have been captured by the leak strength parameter λ. We now discuss this possibility in Lines 350-355 of the revised manuscript.

Lines 276-280. The authors indicate that the fact that choice RTs are shortened on easier trials is indicative of an evidence accumulation process informing decisions. In fact, several prominent models that invoke little (e.g. Urgency Gating) or no (e.g. Extrema) would also predict faster choices when evidence is stronger.

We did not mean to argue that the sequential sampling model (SSM) is the only model that could explain the effect of choice difficulty on RT – we agree that there are other models that may also potentially fit our data. However, our study does not examine the question which specific decision model provides the best fit to our behavioral data, but rather whether and how differences in altruism relate to well-established model-predicted neural evidence accumulation processes that have been documented in several prior studies (Pisauro et al., 2017; Polanía et al., 2014). For the purpose of our study, it thus does not seem important to examine the validity of other choice models that do not allow us to derive these predictions for EEG signals. To prevent any misunderstandings in this respect, we now state explicitly that we do not claim that the OU model is the perfect model for altruistic decision making, but rather frame it as a well-established tool to derive predictions for EEG signals (Lines 1266-1274). Nevertheless, we also stress that we did do model comparison analyses to test the superiority of the winning OU model relative to other variants of sequential sampling models (described in lines 1244-1266 of the revised manuscript). These changes should have removed all ambiguities in this respect.

321-322 – the 'more technical parameters' leak and NDT. Please explain what is meant by 'more technical'.

We apologize for the imperfect wording here. What we meant is that it is more difficult to link these parameters (i.e., leak strength and non-decision time) to specific neural signals underlying sensory, valuation, or decision processes. We have removed the term “more technical parameters” in the current manuscript and now explain that these parameters are less plausible to capture the cognitive or neural mechanisms underlying the valuation or decision processes that are of interest for our question (Lines 344-346).

401-402 – NDT as speed of pre-decision processes – this should be changed to read speed of pre- and post-decision processes as NDT also include motor execution delays.

Thank you for this comment, we agree. We have now re-defined nDT as the speed of pre- and post-decision information processing (Line 440).

Reviewer #2 (Recommendations for the authors):This is an interesting and carefully designed study on altruistic choice that applies rigorous computational modeling methods and observes intriguing correlates of contextual and individual difference influences on behavior. I think it deserves to be published in eLife, and in general think it is a well done paper.

We thank the reviewer for the positive comments and the many fruitful suggestions. We have followed all these suggestions and have performed many additional analyses of our data. Several of these have established clear correspondences between the modelling and the EEG data, and we have added these to the main text. Other exploratory analyses yielded no clear-cut results, but we have included these in Appendix 1 nevertheless. We also briefly describe these exploratory analyses in the main text where appropriate, to give the reader a clearer conceptual understanding of our findings and to pre-emptively answer related questions that future readers of our manuscript may have. We thank the reviewer again for these suggestions, which have helped us to more comprehensively analyze our data from all these angles.

In your introduction, you frame the questions your paper will address around determining whether altruistic choice is accomplished via fundamentally different mechanisms in different contexts and across different individuals. I found myself wondering whether you could make this clearer by emphasizing the distinction between qualitative differences (i.e., process level differences) and merely quantitative differences (i.e., differences in sensitivity of a process to a quantity, but not in the fundamental stages of processing) and whether it might help to have a figure that illustrates the different possibilities and different stages at which one might expect differences.

We apologize if we did not make our research questions clear enough in the original manuscript. We have now revised the relevant statements to avoid further misunderstandings. Specifically, we put more emphasis on the question of whether and how contextual and individual differences in altruism are produced by specific aspects of basic information processing, valuation, or evidence accumulation throughout different temporal stages of the decision (Lines 117-138). We have also rewritten all relevant statements throughout the paper to avoid misunderstandings.

We clarified our hypotheses with clearer reference to previous studies as follows: Since different evidence-accumulation related brain regions have been identified in different types of decision making (e.g., parietal region for perceptual decision making vs. frontal and parietal regions for value-based decision making (Polanía et al., 2014)), decisions reported via the same manual actions may still involve distinct neural evidence accumulation processes in different contexts. Moreover, altruistic decisions driven by different motives, or in people with different altruistic preferences, have also been found to be supported by distinct brain networks (Hein et al., 2016). Therefore, individual and situational differences in altruism may emerge from differences in various neurocognitive processes that unfold during the timecourse of a choice, ranging from early sensory processing to value computation to the final evidence accumulation processes (Harris et al., 2018) (Lines 180-201). We also explain, in line with the reviewer’s suggestion, that these differences may reflect either varying sensitivity of these processes to payoff information or qualitative differences in what processes are involved in the first place. We have now made this distinction between the different possible sources of individual differences much clearer.

Lines 125-126 of the introduction: you point to a question of whether differences reflect involvement of hard-wired choice mechanisms. I'm not sure any of your data can speak to whether this is hard-wired, and this doesn't really seem to be the focus of your analysis anyway, so I would probably suggest avoiding this as a focus in your intro.

We agree with the reviewer that our goal is not identifying whether the mechanisms are “hard-wired”, our choice of wording here was subpar. We have now removed the related statements.

Lines 304-306: could you actually spell out what the winning model was here in the results, to save the lazy reader from having to dig into methods and SI Materials to figure this out?

We have now added a description of the winning model to the Results section (Lines 322-357).

Not critical, but it is interesting to me that in the GLM analysis, you find a significant interaction effect between self and other payoff. That doesn't show up in your O-U model, if I understand correctly, or in any of your subsequent analyses. Is this interaction predicted by Charness-Rabin, or by your EEG results?

Analysis of the behavioral data indeed revealed a three-way interaction between self-payoff change, other-payoff change, and inequality context. The post-hoc analyses showed that for choices that would decrease the other’s payoff, participants reacted less strongly to the associated possible increases in self-payoff in the advantageous context. For choices that would increase the other’s payoff, people were equally sensitive to increases in self-payoff for both inequality contexts.

To examine whether our SSM model can predict this behavioral pattern, we performed a model simulation analysis and showed that for choices that would decrease the other’s payoff, participants should indeed react less strongly to the associated possible increases in self-payoff in the advantageous context (lower slope of Self-payoff Change function in ADV than DIS for Decreased Other-payoff (OP) trials; 1.37 ± 0.06 vs 1.64 ± 0.09, ADV vs DIS: 95% CI [- 0.44 – – 0.09], Cohen’s d = – 0.51, t(37) = -3.14, p = 0.003). For choices that would increase the other’s payoff, subjects should also react less strongly to any associated increase in self-payoff in the advantageous context, but this effect is smaller (Increased Other-payoff (OP); ADV: 1.45 ± 0.07, DIS: 1.73 ± 0.10, ADV vs DIS: 95% CI [- 0.49 – 0.07], Cohen’s d = -0.44, t(37) = -2.69, p = 0.011). Taken together, these results confirm that our SSM model predicts the behavioral pattern observed in the data. We have added this analysis to the manuscript (main text: Line 278 and Appendix 1: section of “Behavior: Choice depends differentially on self- and other-payoffs across contexts”) and explicitly mention this clear correspondence between the model-free and the model-based analysis. Thanks to the reviewer for suggesting this interesting model simulation, which has strengthened our paper.

What are we to make of the fact that individual differences in parameters are strongly correlated across the contexts, but the EEG signals seem to be categorically different? This might be worth explicitly addressing in the Discussion section.

The reviewer’s observation is only partially correct: Only the stimulus-locked ERP results show categorical differences between the contexts, whereas the response-locked ERP results reveal similar signals across contexts, in line with the correlations in DDM parameters across contexts. Thus, the two types of analyses of the processes leading up to the final decision converge on supporting the hypothesis that the decision processes occurring late during the choice process are similar in both contexts. In contrast, only the stimulus-locked ERP signals that index early stimulus processing exhibited categorically different patterns in the two contexts.

This pattern of results is actually not surprising, since SSM model parameters capture the evidence accumulation process, which is more closely related to the ERP signals leading to decisions. In contrast, stimulus-locked ERP signals are more temporally close to early perceptual, attentional, or valuation processing of choice-relevant information, so contextual differences in these signals reflect how the neural processes related to extraction of choice-relevant information from the stimuli differ between the two inequality contexts. Thus, our results coherently suggest that while the computation/implementation of early processing of choice-relevant information differs, the computation/implementation of the late evidence accumulation process does not differ across contexts. In the revised manuscript, we have followed the reviewer’s suggestion and now explicitly discuss how the findings are congruent with the overall picture of similarities in late decision mechanism, but profound differences in early stimulus processing (p.44). This should help to avoid any ambiguities or misunderstandings.

Was there any correlate of the difference in decision bound between ADV and DIS contexts across participants? In Harris et al. (2018), a correlate of decision bound was observed in response-locked data. I found myself wondering whether you might observe something similar here.

We followed both reviewers’ and the editor’s suggestions to check the correspondence between decision bounds and response-locked EEG signals. Specifically, we re-analyzed the response-locked ERP signals and found that the ERP amplitude at response (in the time window of 100 to 0 ms before response) in the DIS context is indeed higher than that in the ADV context. This is entirely consistent with our sequential sampling model result showing higher decision thresholds in DIS than ADV context. We have now included this analysis in the Results section (Lines 396-400 and Figure 2 – —figure supplement 1).

However, the correlations between decision threshold and ERP amplitude at response were close to 0 in both contexts (r(DIS) = -0.04, p = 0.80; r(ADV) = -0.05, p = 0.77). Such a discrepancy between the current result and the results in Harris et al. (2018) may be due to two reasons. First, the time window we focused on is 100 to 0 ms before response, which is different from the time window of 300 to 180 ms before response as in Harris et al. (2018). In Harris et al. (2018), ERP signals in the window from 300 to 180 ms before response were first shown to vary across contexts and the neural signals in this time window were then input into a DDM to account for the variance in decision threshold, but such contextual differences in ERP signal were not observed in our response-locked analyses. Therefore, we only focused on the time window right before response in the current study. In our analyses, we did not include the ERP signals into the model estimation but correlated ERP signals with the decision threshold. Therefore, in our study, greater noise may have buried the contribution of ERP signals to the decision threshold in the sequential sampling model. We have not added these null results to the manuscript, but would be happy to do so if the reviewer or editor think this is necessary or helpful.

Similarly, was there any correlate of the starting bias parameter? I am generally not a huge fan of DDM models in which the decision bounds do not actually map on to the responses subjects make, as you do here with your equal/unequal choice bounds, because they imply that subjects somehow precognitively know whether the first or second option will be the more equal and set their starting bias accordingly. Conceptually this makes no sense, and I am on record (Teoh et al., 2020) as arguing that this starting bias could result from post-stimulus attention and value accumulation mechanisms rather than pre-stimulus biases. Acknowledging that you find a very tiny BIC advantage for your model over first-second option bound model (though it's not clear to me whether that small difference could arise due to chance), and that the results end up being pretty similar across the two models for the parameter you focus on (i.e., omega), I found myself wondering whether the EEG signals might be useful here. For example, if you looked for an EEG correlate of starting bias in the period just before presentation of the second option, and correlated that with starting bias (either in the equal/unequal or first/second bound models), this might be another way of bolstering your message that most of the computations that influence choices/individual differences happen not at the late accumulation stage, but at the early perception and valuation stages.

We agree that it would be interesting if we could relate EEG signals to the starting point bias. Following the reviewer’s suggestion, we re-analyzed the ERP responses time-locked to the presentation of the second option, by taking 500 to 200 ms before the stimuli onset as the baseline period to re-perform the baseline correction. We then conducted the GLM analyses (i.e., one GLM model with the value of the self- and other-payoffs in the first option as regressors and one GLM model with the difference between self- and other-payoffs in the first option as the regressor to predict ERP signals) over 200 to 0 ms before the second option onset.

We correlated the starting point bias in either the winning OU model or the OU order model with the neural signals related to the self-payoff, other-payoff, or the self- vs other-payoff difference in the first option, or the raw ERP signals in either context. Unfortunately, these analyses did not reveal any significant clusters in which the pre-stimuli ERP signals were related to the starting point bias across participants. These analyses are now included in the main text (Lines 595-601) and Appendix 1 for the benefit of future readers.

Can you say more about what it means that the ERP-behavior correlation with δ-other in the DIS context is observed from 320-440ms stimulus-locked, but the phase coupling correlation is observed in response-locked data from 520-460ms prior to choice? Do these periods actually overlap? How do you see these signals as related if they occur in different time periods?

These two time windows overlap, as we now show explicitly in the new Results section (Lines 709-723). We extracted the starting and end time points of the time window of ~320 to 400 ms after stimulus onset and defined the temporal intervals between each of these two time points and the response time point as the transferred response-locked time points for each trial. We then averaged these transferred response-locked time points for each participant. The results showed that the time window for ~320 to 400 ms after stimulus onset corresponds to a response-locked time window of ~530 (95% CI [600 – 470]) to 450 (95% CI [520 – 390]) ms before response, which contains the time window ~ 520 to 460 ms before response reported in our manuscript. This makes it likely that the two effects reflect overlapping mechanisms, a point that we now discuss on p.32.

Is there any signal that correlates with overall integrated value/relative value of the second option? Fronto-central signals are often seen in studies of food choice from between 400-500ms after stimulus onset, for instance. Is something similar observed here during presentation of the second option? If so, how does that signal compare spatially and temporally to the signals related to δ-self and δ-other that you see here?

We ran similar GLM analyses of ERP signals related to either the integrated value (utility difference between chosen and unchosen option) or the relative value of the second vs first option separately for DIS and ADV contexts. For the integrated value, we did neither observe any significant cluster within each context nor any cluster showing differences between the two contexts. These null results are not surprising since the integrated values are commonly shown to be encoded in the ventromedial prefrontal cortex (Bartra et al., 2013), which may not be easily detected by EEG. These analyses are included in the main text (Lines 595-601) and Appendix 1 for the benefit of future readers.

Since the relative value of the second relative to the first option is highly correlated with the difference in self-payoff between the two options, we observed ERP signals that are very similar to those we identified in the analyses of self-payoff change in the main analyses. To avoid double dipping into the data, we thus do not report these as a separate analysis but would be happy to do so with the appropriate caveats if the editor or reviewer think this is necessary or helpful.

One of the most interesting or unexpected findings, to me at least, is the lack of any consistent signals for self and other across the two contexts, at least during presentation of the second option. While you acknowledge in your Discussion section that some of this may have to do with your experimental design, I found myself wondering whether you could leverage your design a bit more to provide greater insight into how this might play out, computationally. For example, the wheel-and-spoke design of your payoffs (Figure 1-supplement 1) means that, once a person knows the values of the first option for self and other, they also known within a narrow range what the values of the second option can possibly be. While it is only this relative value that matters for choice, it is clear that something occurring in response to the first option seems to 'set' the way that people process the second option. This could be what drives differences in starting point bias, as well as differences in response to δ-self/δ-other. You don't report any analyses performed during the time period of the first option, but I found myself wondering whether this might not be an extremely interesting time period to explore. For instance, do you see any signal that correlates with the overall magnitude of self or other during this period, time-locked to stimulus onset? Do you see a signal that differentiates ADV from DIS inequality contexts? Does that signal then predict on an individual basis how people respond to presentation of the second option? You already have a lot of analyses packed into this paper, so I'm not sure exactly that I am suggesting you include a whole new section on the first stage of your paradigm in this paper, but to be honest I found myself wondering whether that dynamic interaction between first and second-stage calculations might not end up being a more interesting and revealing place to explore, and yield a more interesting message about what predisposes people to act generously or selfishly, or to attend to self or other outcomes.

We did not report any results of the ERP signals during the presentation of the first option as we did not observe any significant ERP responses related to self-/other-payoff or self- vs other-payoff difference of the first option in our original analyses. Nevertheless, we followed the reviewer’s suggestion and re-analyzed ERP signals time locked to the presentation of the first option. We performed GLM analyses in the same manner as we did in the main analyses, by setting up one GLM with the value of the self- and other-payoffs as regressors and one GLM with the difference between self- and other-payoffs as the regressor to predict ERP signals.

We found the following. First, there were no significant ERP clusters related to the magnitude of self-/other-payoff or self- vs other-payoff difference in either inequality context. Second, there were no significant ERP clusters differently associated with self-/other-payoff or self- vs other-payoff differences across the two contexts. Third, there were no significant ERP clusters in which the responses to self-/other-payoff or self- vs other-payoff difference are related to the starting point bias in either context. These analyses are included in the main text (Lines 595-601) and Appendix 1 for the benefit of future readers.

If I understood correctly, when you look at individual difference correlates, you only look at correlates of omega within each context separately. But it seems to me that another way of looking at this is via two separate individual differences: difference in overall omega (averaged over the two contexts) and differences in change in omega across the two contexts. I would be particularly interested in whether any of the changes in the signals you observe across context related to δ-self are related to omega-difference.

We agree with the reviewer that it would be interesting to observe a systematic relationship between ERP signals and the weight on others’ payoff. We re-analyzed our EEG data by (1) correlating averaged weight on others’ payoff (ω) with ERP signals related to self-payoff change in each context, and (2) correlating the difference in ω between contexts with the difference in ERP signals related self-payoff change between contexts. However, these analyses revealed no significant cluster. This analysis is included in the main text (Lines 678-682) and Appendix 1 for the benefit of future readers.

Is it possible to do a similar dWLPI analysis for the sensors that were differentially sensitive to δ-self in the two contexts? One might speculate that if connectivity between perceptual/value signals and evidence accumulation processes is a key mechanism for information transfer, then one should see an analogous result when looking at these sensors. Especially given that you don't observe any amplitude differences that correlate with omega, it might be that such differences are more likely to arise from information transfer differences.

We thank the reviewer for this suggestion, but unfortunately, since the sensors involved in evidence accumulation largely overlap spatially with those involved in processing of self-payoff changes (see Figure 2C & 2D leftmost panel and Figure 4A & 4C left panel), we cannot provide unambiguous evidence for the dWPLI strength between the overlapping sensors.

On a similar vein: you find that both ERP sensitivity to δ-other and dWLPI predict differences in generosity/differences in omega in the DIS context. Is one of these more important, or mediating the other? For example, if information contained in ERPs is being transmitted via neural coupling, one might expect that the dWLPI either mediates or moderates the influence on choice of individual differences in ERP sensitivity.

We found that there was no significant correlation between the ERP responses to other-payoff change (320 to 400 ms after second option onset) and frontal-parietal dWPLI strength (520 to 460 ms before response) across participants (Pearson r = 0.18, p = 0.28; Kendall tau = 0.001, p = 1.00). Moreover, the mediation model (with the ERP signal as the independent variable, dWPLI strength as the mediator variable, and decision weight as the dependent variable) could not be established as the path from ERP responses to dWPLI strength; the mediation model as a whole was also not significant. Therefore, we chose not to speculate about the relationship between ERP responses and dWPLI strength. We have nevertheless included these analyses in the current main text (Lines 759-762) and Appendix 1 for the benefit of future readers.

In lines 813-818, you suggest that early signals (240-360ms post-stimulus) related to inequality processing, and that later signals (440-800) relate to the conflict between inequality level and self-/other-payoff change. I couldn't quite tell whether you are trying to argue that these signals are qualitatively different or not, and if so, on what basis you conclude that. The signals look fairly similar to me in terms of what they are sensitive to, so it wasn't obvious to me that one can conclude they are doing different things. Can you clarify?

We agree with the reviewer that the signals in these two time windows appear very similar. We, therefore, did not mean to argue that they reflect qualitatively different processes. In the revised manuscript, we have removed the ambiguous wording. Now, we suggest that these signals both reflect the integrated processing of self-payoff with inequality-related contextual information.

In lines 834-837 of the conclusion, you suggest that the different temporal windows of sensitivity to self payoff change might account for the differences in reaction times. But your modeling results suggest that much of this difference could be due to changes in threshold. This probably deserves acknowledgement.

We now acknowledge this point in the Discussion section.